# Multiproxy analysis unwraps origin and fabrication biographies of Sardinian figurines: On the trail of metal-driven interaction and mixing practices in the early first millennium BCE

Daniel Berger[1,☉,*], Valentina Matta[2,‡], Nicola Ialongo[3,‡], Heide W. Nørgaard[4,‡], Gianfranca Salis[5], Michael Brauns[1,‡], Mads K. Holst[4], Helle Vandkilde[3,‡,*]

1 Curt-Engelhorn-Zentrum Archäometrie gGmbH, Mannheim, Germany, 2 Graduate School, Faculty of Arts, Aarhus University, Aarhus Centrum, Denmark, 3 Department of Archaeology and Heritage Studies, School of Culture and Society, Aarhus University, Højbjerg, Denmark, 4 Moesgaard Museum, Højbjerg, Denmark, 5 Soprintendenza Archeologia, Belle Arti e Paesaggio per la Città Metropolitana di Cagliari e le Province di Oristano e Sud Sardegna (Sabap, Mibact), Cagliari, Italy

☉ These authors contributed equally to this work.
‡ These authors also contributed equally to this work.
* daniel.berger@ceza.de (DB); farkhv@cas.au.dk (HV)

## Abstract

This article presents a multiproxy investigation of metal samples obtained from 48 Nuragic figurines (so-called *bronzetti*) and three copper bun ingots. These objects originate from three prominent Sardinian sanctuaries and one unidentified site, dating to the late Nuragic period of the early first millennium BCE. The dataset significantly expands the existing scientific database and unwraps the complex fabrication biographies of the figurines from ore to finished object. The investigation employs an advanced archaeometallurgical approach, integrating conventional trace-elemental and lead isotope analyses with rarely used copper, tin, and osmium isotope measurements. This methodological combination allows for a more reliable identification of the original metal sources used in the production of the objects, namely copper from the Iglesiente-Sulcis district in southwest Sardinia, with the Sa Duchessa mine as the most likely supplier, in addition to copper from the Alcudia valley or the Linares district in the Iberian Peninsula. Notably, the combination of analytical proxies reveals the mixing of copper from these distinct regions, while ruling out the exploitation of Sardinian tin resources. Furthermore, the osmium isotope ratios confirm the use of Sardinian copper and exclude the alloying of local lead with imported copper. These results shed light on local metallurgical practices and distribution strategies in Nuragic Sardinia, but also on Sardinia's broader role and position in the Mediterranean world during the transition from the Late Bronze Age to the Early Iron Age.

**Data availability statement:** All relevant data are within the paper and its Supporting Information files.

**Funding:** Grant agreement 23–1869 to HV, MKH, GS. Augustinus Foundation funding the Metals & Giants project. https://augustinus-fonden.dk/en The foundation played no role in the study design, data collection and analysis, decision to publish, or preparation of the manuscript.

**Competing interests:** The authors have declared that no competing interests exist.

## 1. Introduction

The metallurgy of the Nuragic culture and the origins of the copper during the Final Bronze Age of Sardinia at the threshold to the Iron Age, c. 1000–700 BCE, remain insufficiently understood. Limitations in existing knowledge complicate understanding the role of Sardinian metallurgy, both locally and within the broader Mediterranean–Atlantic network. The island is often regarded as a peripheral player among metal-producing Bronze Age societies, a recipient rather than a producer and transmitter of metals. This inclination to underplay Sardinia's role is further reflected in arguments emphasising the high impact – whether direct or indirect – of external partners such as Cyprus, Iberia, northern Italy, and the Levant [1–5].

Recent discussions of Sardinian metallurgy typically engage with the provenance of copper and tin, but also with the question of whether Sardinia's local metal reserves, especially of copper, were exploited at all during the Bronze and Iron Age [2,6–9]. Previous studies have provided insights into the elemental and isotopic systematics, as well as the alloying practices, of Nuragic copper-based artefacts. However, the debate over whether Sardinian metalwork relied partially or entirely on the island's own copper resources is still unresolved [6,10–17]. Given the island's wealth of relevant metal sources (Fig 1), this question cannot simply be ignored. Nevertheless, sourcing Sardinian copper and tin has proven challenging for several reasons. One of these is the significant overlap of Sardinia's lead isotope ratios with those of other relevant copper-bearing regions in the Iberian Peninsula and the Arabah valley, making a clear distinction between them often difficult. Another complication arises from the frequent presence of elevated lead contents in final products from Sardinia, potentially indicating intentional additions that can obscure the identification of the original sources of copper. Finally, a persistent issue is the substantial presence of Cypriot copper [20–26]. Copper from Cyprus reached Sardinia in the form of oxhide ingots mainly between c. 1300 and 1100 BCE, and kept circulating in fragment form into the early first millennium [3,6,27,28]. However, the extent to which Cypriot copper was actually utilised in local metallurgy is uncertain, in part due to conflicting isotopic and chemical signatures between locally produced Sardinian metalwork and oxhide ingots. Overall, the entire subject area is open for fresh inquiry.

One way to tackle the subject is to investigate metal objects of undisputed local production. The *bronzetti nuragici* or small bronze figurines (Fig 2), iconic symbols of Sardinia and a hallmark of its vibrant metalworking tradition, are such objects. These figurines represent diverse facets of Nuragic society, including the so-called 'chieftains', warriors, offerers, animals, and boats. Scholars have identified two groups of *bronzetti*: the earlier and more prominent Uta-Abini figurines (c. 950–800 BCE) and the later, rather marginal *Mediterraneizzante* figurines (c. 800–700 BCE). These two types may represent specific social groups or workshops [29–33], but such a distinction relies almost exclusively on stylistic criteria. It has been suggested that the bronze statuettes were commissioned by the Nuragic elite for use as votive offerings, with the dual intention of displaying elite status and claiming ancestral belonging to the epic '*bronzetti* community' [29,31]. Eastern Mediterranean influence on the

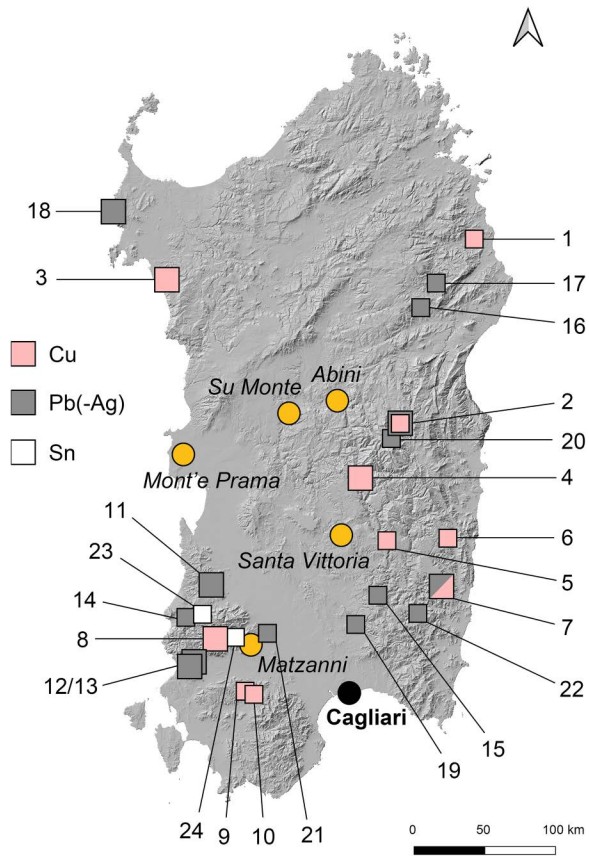

**Fig 1. Ore mineralisation of copper (Cu), tin (Sn), and lead-silver (Pb-Ag) in Sardinia.** Larger symbols were the most relevant for prehistoric people, while mineralisation or occurrences denoted with smaller symbols were either not accessible for the ancient miners or of insignificant size (assessment after [18,19]). 1: Val Barisone, 2: Correboi, 3: Calabona, 4: Funtana Raminosa, 5: Monte Nieddu, 6: Baccu Talentinu/Bau Arenas, 7: Baccu Locci, 8: Sa Duchessa – Domusnovas, 9: Rosas – Sa Marchesa, 10: Barisonas, 11: Montevecchio, 12: Monteponi, 13: San Giovanni, 14: S'Oreri/Santa Lucia, 15: Genna Tres Montis, 16: Sos Enattos, 17: Guzzurra, 18: Argentiera, 19: S'Ortu Becciu, 20: Gennargentu, 21: Monte Zippiri, 22: 'Sarrabus siver lode', 23: Perdu Cara, 24: Canale Serci (map: V Matta/D Berger using Open Source QGIS software 3.20 Odense and geo reference data from https://www.sardegnageoportale.it/navigatori/sardegnamappe/ both under a CC BY 4.0 license).

iconography has been a long-standing controversy, but recent scholarship proposes that the *bronzetti* reflect broader pan-European Bronze and Early Iron Age traditions and connectivity [5,34–39].

*Bronzetti* are found predominantly in sanctuaries across Sardinia (Fig 3), especially at water temples, and only rarely in burials and settlements. Scholars differ on the dating of the earlier Uta-Abini group, with some placing them in the Final Bronze Age (1200–900 BCE), others in the Sardinian Early Iron Age (950–800 BCE) [14,29,30,36,40]. Securely datable contexts for these objects in Sardinia are scarce, making chronology heavily reliant on findings from the Italian mainland. A small number have been discovered in well-documented contexts within the Villanovan culture of central Italy that can be reliably dated to the Early Iron Age (ca. 950–800 BCE) [e.g., 41,42]. This chronology is followed here.

In the 1970s and 1980s, the first chemical analyses were conducted of Nuragic *bronzetti* from collections outside Sardinia, using atomic absorption spectrometry (AAS) [11,43,44]. Shortly thereafter, investigations expanded to include figurines from Sardinian collections. Angelini and colleagues [13,45] published analytical data of Uta-Abini style *bronzetti* from the Nuragic sanctuary of Abini-Teti and from another unidentified location, employing both AAS and lead isotope analysis. Notably, the lead isotope composition matched minerals in the 'metalliferous ring' of southwest Sardinia [7]. However, as

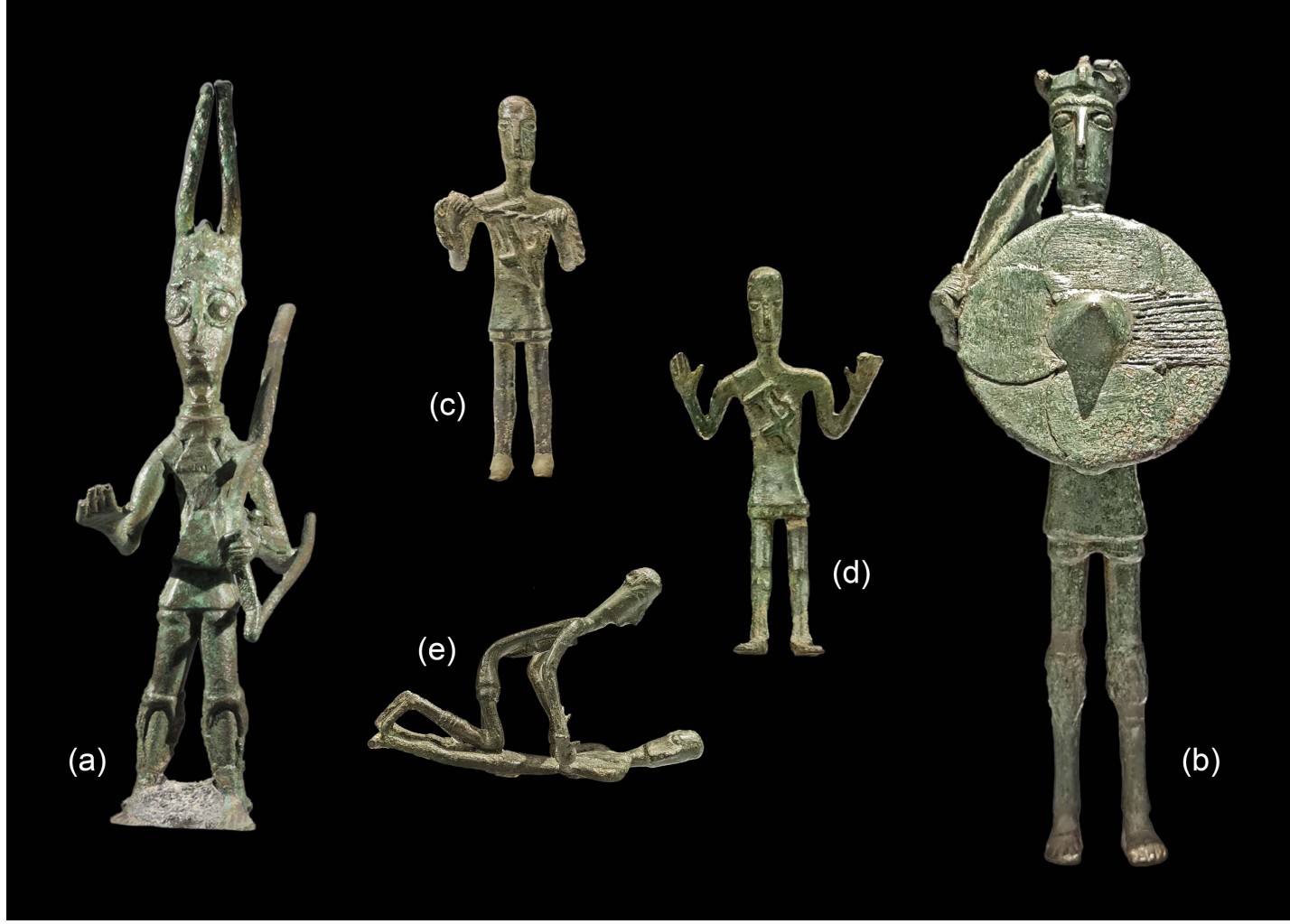

**Fig 2. Bronzetti of the Uta-Abini style from Sardinia.** (a) Abini-Teti, (b–e) Monte Arcosu-Uta; images not to scale (photos: HW Nørgaard, D Berger).

only isotopic ranges were provided, rather than numerical values, this result remains only partially assessable. In recent years, numerous chemical analyses have been performed on *bronzetti* of both types, while only one lead isotope analysis has been added [14,17,19,42]. This is in stark contrast to the extensive lead isotope datasets available for other copper-based metalwork and ingots from Sardinia [1,6–8,15,46,47]. Recent advancements in archaeometallurgy, including progress in analytical techniques, provide evidence for foreign metal sources in Sardinian metal production from beyond Cyprus, such as Iberia and Timna in the Arabah valley [8,15,48,49]. This supports the hypothesis that Sardinia imported metal from various regions. However, it does not exclude that the abundant local copper resources were exploited (Fig 1).

The present study aims to contribute new knowledge to this aspect through a multiproxy approach combining chemical and isotopic analysis of about fifty samples from three Nuragic sanctuaries and a further, unidentified Nuragic site. As such, the approach builds on previous studies that analysed *bronzetti* non-destructively and chemically, but which offered limited insights into raw material supply [11,14,16,17]. By sampling the figurines and employing integrated chemical and isotopic analyses, including innovative techniques using tin, copper, and notably osmium isotopes, the study significantly expands the database to a hitherto unprecedented level. In consequence, the findings provide fresh insights into

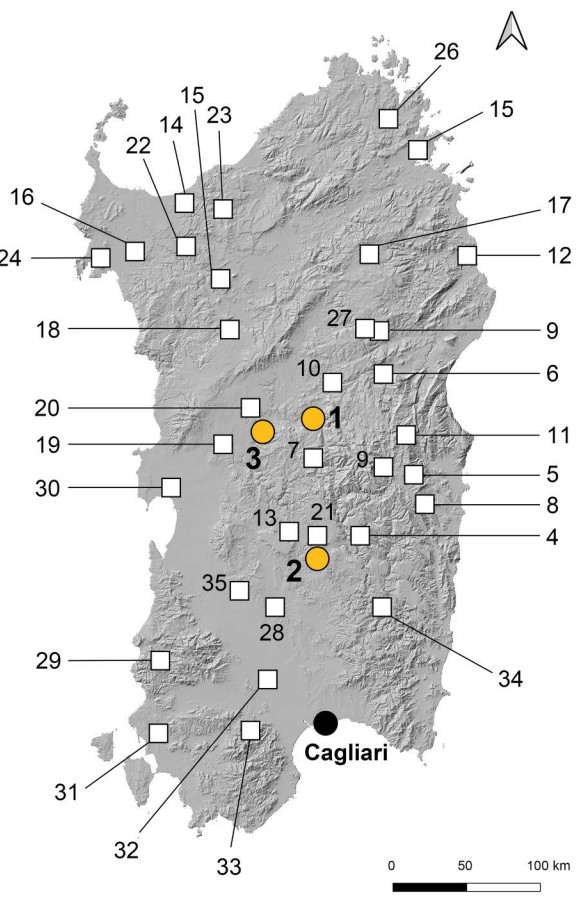

**Fig 3. Localities of the bronzetti analysed in the present study (1–3) and figurines from other sites (4–35) (without claiming completeness).** 1: Abini-Teti, 2: Santa Vittoria-Serri, 3: Su Monte-Sorradile, 4: Domu de Orgia-Esterzili, 5: S'Arcu 'e is Forros-Villagrande Strisaili, 6: Sa Sedda 'e sos Carros, 7: Sórgono, 8: Lanusei, 9: Su Tempiesu-Orune, 10: Nurdole-Orani, 11: Urzulei-Sa Domu de S'Orku, 12: Nuraghe Pizzinu-Posada, 13: La Rotonda-Genoni, 14: Serra Niedda-Sorso, 15: Monte S. Antonio-Siligo, 16: Camposanto-Olmedo, 17: Su Pedrighinosu-Ala dei Sardi, 18: Mulino-Bonorva, 19: Santa Cristina-Paulilatino, 20: Aidomaggiore, 21: Adòni-Villanova Tulo, 22: Sa Mandra 'e sa Giua-Ossi, 23: Nuraghe Orku-Nulvi, 24: Flumenelongu-Alghero, 25: Cabu Abbas-Riu Mulinu, 26: Nuraghe Albúcciu-Arzachena, 27: Santa Lulla-Orune, 28: Nuraghe Cummossariu-Furtei, 29: Antas, 30: Mont'e Prama, 31: Monte Sirai, 32: Decimoputzu, 33: Monte Arcosu-Uta, 34: Funtana Coberta-Ballao, 35: Sardara (map: V Matta/D Berger after [36] using Open Source QGIS software 3.20 Odense and geo reference data from https://www.sardegnageoportale.it/navigatori/sardegnamappe/ both under a CC BY 4.0 license).

Sardinia's role in the metal trading networks of the Bronze and Iron Age, and also into the origins of the materials used in the production of its iconic *bronzetti*. Ultimately, the study also opens new avenues for distinguishing copper source regions with overlapping lead isotopic and chemical signatures.

## 2. Materials and methods

Most of the studied figurines originate from three key Sardinian sanctuary sites – Su Monte-Sorradile, Abini-Teti, and Santa Vittoria-Serri (Fig 3). Nuragic sanctuaries were political and religious hubs in which large amounts of goods – especially metalwork – were dedicated. Offerings include ingots, hack metal, tools, ornaments, weapons, and votive items, and there is also evidence for active metalworking at several sites [50–52]. Su Monte features a unique, monumental circular building with a stone-sculpted basin used for rituals [51], while Santa Vittoria, a sprawling sanctuary with numerous huts and temples, has produced a significant number of bronze figurines and other metal artefacts [53]. Abini is particularly

notable for its large collection of *bronzetti*, including statuettes of warriors, archers, and boats, as well as indications of metallurgical activities [54,55]. This evidence suggests that Nuragic sanctuaries could have played a pivotal role in Sardinia's metallurgical and copper distribution networks during the Bronze and Iron Ages [52].

A total of 48 figurines from Santa Vittoria (n = 31), Su Monte (n = 8), Abini (n = 3), and one unknown site (n = 6) from the collection of the National Archaeological Museum in Cagliari were sampled. The majority of samples derived from fragments of *bronzetti* and represented hands and arms, feet, or horns on helmets (Fig 4). Three additional samples originated from copper bun ingots (Table 1). Given stylistic details, as well as technical and formal peculiarities of the extremities (e.g., different dimensions, execution of stylistic characteristics), there are no two fragments that belong to the same figurine, so that in fact 48 different *bronzetti* were sampled.

The samples were extracted using a jigsaw and then ground to obtain a flat surface for chemical analysis with energy-dispersive X-ray fluorescence spectrometry (EDXRF) (all values in mass% throughout the study). Corrosion products were removed as thoroughly as possible; nevertheless, some samples were found to be heavily weathered, with corrosion products penetrating deeply into the metal core or entirely substituting it. Because of the risk of alteration by the weathering process, such badly preserved specimens were excluded from the more sensitive chemical analysis with quadrupole mass spectrometry with inductively coupled plasma ionisation (ICP-Q-MS) and from copper and tin isotope analysis with multi-collector ICP-MS (MC-ICP-MS). All other samples were subjected to stable copper and tin isotope analysis. Regardless of the state of preservation, all specimens were analysed for their lead isotope composition, while a set of seven specimens was additionally analysed for the osmium isotope composition (Table 1). Details of the sample preparation and the analytical procedures can be found in the S1 Appendix.

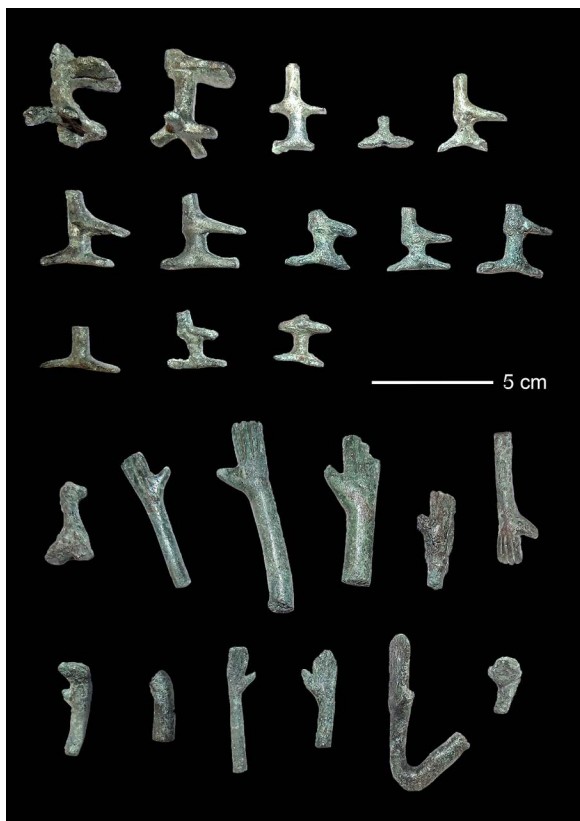

**Fig 4. Selection of sampled object parts from the analysed bronzetti.** The top three panels feature feet and legs, partly with remnants of the casting sprues, the bottom two panels display hands and arms (photos: HW Nørgaard).

**Table 1.  Artefacts analysed in the study and analyses carried out. EDXRF – energy-dispersive X-ray fluorescence spectrometry; ICP-Q-MS – quadrupole mass spectrometry with inductively coupled plasma ionisation; LIA – lead isotope analysis; TIA – tin isotope analysis; CIA – copper isotope analysis; OIA – osmium isotope analysis.**

| Lab. no. | Orig. no. | Find spot | Object, sampled part | Analysis |
|---|---|---|---|---|
| MA-202494 | A1401 | Su Monte (Sorradile) | Bun ingot | EDXRF, ICP-Q-MS, LIA, CIA, OIA |
| MA-202495 | A1798 | Su Monte (Sorradile) | Bun ingot | EDXRF, ICP-Q-MS, LIA, CIA |
| MA-202496 | A1067 | Su Monte (Sorradile) | Bun ingot | EDXRF, ICP-Q-MS, LIA, CIA |
| MA-202498 | A921 | Su Monte (Sorradile) | *Bronzetto*, arm | EDXRF, LIA |
| MA-202501 | A2295 | Su Monte (Sorradile) | *Bronzetto*, right foot and both legs | EDXRF, ICP-Q-MS, LIA, TIA, CIA |
| MA-202502 | A1864 | Su Monte (Sorradile) | *Bronzetto*, arm | EDXRF, ICP-Q-MS, LIA, TIA, CIA |
| MA-202503 | C730/1 | Su Monte (Sorradile) | *Bronzetto*, foot | EDXRF, ICP-Q-MS, LIA, TIA, CIA |
| MA-202504 | C2958 | Su Monte (Sorradile) | *Bronzetto*, horn? | EDXRF, ICP-Q-MS, LIA, TIA, CIA |
| MA-202505 | C938 | Su Monte (Sorradile) | *Bronzetto*, under the foot (lead) | EDXRF, LIA |
| MA-202506 | C1838 | Su Monte (Sorradile) | *Bronzetto* | EDXRF, ICP-Q-MS, LIA, TIA, CIA |
| MA-202507 | C2214 | Su Monte (Sorradile) | *Bronzetto*, arm | EDXRF, ICP-Q-MS, LIA, TIA, CIA |
| MA-202508 | C3000 | Su Monte (Sorradile) | *Bronzetto*, arm | EDXRF, ICP-Q-MS, LIA, TIA, CIA |
| MA-202499 | B467 | Abini (Teti) | *Bronzetto*, left hand and arm with stick? | EDXRF, LIA, TIA, CIA |
| MA-202500 | B468 | Abini (Teti) | *Bronzetto* | EDXRF, ICP-Q-MS, LIA, TIA, CIA |
| MA-202509 | B214 | Abini (Teti) | *Bronzetto*, legs? | EDXRF, LIA, TIA, CIA |
| MA-202497 | A1346 | unknown site | *Bronzetto*, foot | EDXRF, ICP-Q-MS, LIA, TIA, CIA |
| MA-202510 | 2566–2 | unknown site | *Bronzetto*, hand and arm | EDXRF, LIA |
| MA-202511 | 2566–4 | unknown site | *Bronzetto*, foot | EDXRF, ICP-Q-MS, LIA, TIA, CIA |
| MA-202512 | 2566–5 | unknown site | *Bronzetto*, hand and arm | EDXRF, LIA |
| MA-202513 | 2566–1 | unknown site | *Bronzetto*, foot | EDXRF, ICP-Q-MS, LIA, TIA, CIA |
| MA-202514 | 2566–3 | unknown site | *Bronzetto*, hand and arm | EDXRF, ICP-Q-MS, LIA, CIA |
| MA-202515 | A1472–43221a | Santa Vittoria (Serri) | *Bronzetto*, foot | EDXRF, ICP-Q-MS, LIA, TIA, CIA, OIA |
| MA-202516 | A1472–43221b | Santa Vittoria (Serri) | *Bronzetto*, foot | EDXRF, LIA |
| MA-202517 | A1472–43221c | Santa Vittoria (Serri) | *Bronzetto*, foot | EDXRF, ICP-Q-MS, LIA, TIA, CIA |
| MA-202518 | A1472–43221d | Santa Vittoria (Serri) | *Bronzetto*, foot | EDXRF, LIA |
| MA-202519 | A1472–43221e | Santa Vittoria (Serri) | *Bronzetto*, foot | EDXRF, ICP-Q-MS, LIA, TIA, CIA |
| MA-202520 | A1472–43221f | Santa Vittoria (Serri) | *Bronzetto*, foot | EDXRF, LIA |
| MA-202521 | A1472–43221g | Santa Vittoria (Serri) | *Bronzetto*, foot | EDXRF, ICP-Q-MS, LIA, TIA, CIA, OIA |
| MA-202522 | A1472–43221t | Santa Vittoria (Serri) | *Bronzetto*, foot | EDXRF, ICP-Q-MS, LIA, TIA, CIA |
| MA-202523 | A1472–43221p | Santa Vittoria (Serri) | *Bronzetto*, hand and arm | EDXRF, ICP-Q-MS, LIA, TIA, CIA |
| MA-202524 | A1472–43221r | Santa Vittoria (Serri) | *Bronzetto*, foot | EDXRF, LIA |
| MA-202525 | A1472–43221l | Santa Vittoria (Serri) | *Bronzetto*, hand and arm | EDXRF, ICP-Q-MS, LIA, TIA, CIA, OIA |
| MA-202526 | A1472–43221s | Santa Vittoria (Serri) | *Bronzetto*, foot | EDXRF, LIA |
| MA-202527 | A1472–43221o | Santa Vittoria (Serri) | *Bronzetto*, hand and arm | EDXRF, LIA |
| MA-202528 | A1472–43221i | Santa Vittoria (Serri) | *Bronzetto*, hand and arm | EDXRF, LIA |
| MA-202529 | A1472–43221n | Santa Vittoria (Serri) | *Bronzetto*, hand and arm | EDXRF, LIA |
| MA-202530 | A1472–43221h | Santa Vittoria (Serri) | *Bronzetto*, hand and arm | EDXRF, LIA |
| MA-202531 | A1472–43221m | Santa Vittoria (Serri) | *Bronzetto*, hand and arm | EDXRF, LIA |
| MA-202532 | 43024 | Santa Vittoria (Serri) | *Bronzetto*, foot | EDXRF, LIA |
| MA-202533 | 43025 | Santa Vittoria (Serri) | *Bronzetto*, foot | EDXRF, ICP-Q-MS, LIA, TIA, CIA, OIA |
| MA-202534 | 43027 | Santa Vittoria (Serri) | *Bronzetto*, foot | EDXRF, LIA |
| MA-202535 | 43031 | Santa Vittoria (Serri) | *Bronzetto*, left hand and arm | EDXRF, LIA |
| MA-202536 | 43032 | Santa Vittoria (Serri) | *Bronzetto*, left hand and arm | EDXRF, LIA |
| MA-202537 | 43033 | Santa Vittoria (Serri) | *Bronzetto*, right hand and arm | EDXRF, LIA |

*(Continued)*

**Table 1.** (Continued)

| Lab. no. | Orig. no. | Find spot | Object, sampled part | Analysis |
|---|---|---|---|---|
| MA-202538 | 43034 | Santa Vittoria (Serri) | *Bronzetto*, left hand and arm | EDXRF, LIA |
| MA-202539 | 43035 | Santa Vittoria (Serri) | *Bronzetto*, right hand and arm | EDXRF, ICP-Q-MS, LIA, TIA, CIA, OIA |
| MA-202540 | 43036 | Santa Vittoria (Serri) | *Bronzetto*, right hand and arm | EDXRF, LIA |
| MA-202541 | 43037 | Santa Vittoria (Serri) | *Bronzetto*, right hand and arm | EDXRF, ICP-Q-MS, LIA, TIA, CIA |
| MA-202542 | 43038 | Santa Vittoria (Serri) | *Bronzetto*, right hand and arm | EDXRF, ICP-Q-MS, LIA, TIA, CIA |
| MA-202543 | 43039 | Santa Vittoria (Serri) | *Bronzetto*, right hand and arm | EDXRF, LIA |
| MA-202544 | 43040 | Santa Vittoria (Serri) | *Bronzetto*, right hand and arm | EDXRF, ICP-Q-MS, LIA, TIA, CIA |
| MA-202545 | 43042 | Santa Vittoria (Serri) | *Bronzetto*, left hand, arm and elbow | EDXRF, LIA |

## 3. Results

### 3.1. The elemental composition

The 31 fragments of the figurines from Santa Vittoria are made of bronze, with tin (Sn) contents ranging from 6 to 16% (Table 2). The latter tin concentration of MA-202543 is certainly affected by its highly corroded condition, and thus represents an enriched value compared to the original, uncorroded bronze. The same applies to several highly corroded statuettes from Su Monte, Abini, and the unknown site with moderate or rather high tin values (MA-202498 and MA-202512), but, generally, the 17 figurines from these places display a similarly large tin spread of 3.8 to 17.5%.

Lead concentrations are also highly variable throughout, ranging from 0.07 to 11.3%, though the majority has values around or below 2%. Two samples from Santa Vittoria and one from Su Monte with high lead contents of 5.2, 5.6 and 3.9% (MA-202529, −33 and −01) likely represent intentionally leaded bronzes rather than binary copper-tin alloys, while the 11.3% lead of another figurine from Su Monte (MA-202504) is definitely deliberate. In all other cases with lower lead, the situation is less clear. This is because lead contents can depend on a variety of factors, such as the type of objects or their dating, the paragenesis of mineral species in ore deposits, and not least the potential mixing or recycling of ores and metals. For example, lead can enter the copper unintendedly or unnoticed during the smelting of naturally associated copper and lead ores – which are common in Sardinia [1,56]. Pernicka [56] considers about 5% lead as a threshold to decide whether lead was intentionally added or accidentally introduced alongside the copper. However, he emphasises that this is a non-rigid threshold. Other scholars propose even lower limits of only 2% [e.g., 17].

In addition to tin and lead, the Santa Vittoria figurines contain further elements at trace level (Table 2): silver (Ag) concentrations range from 0.012 to 0.13%, antimony (Sb) from 0.015 to 0.13%, arsenic (As) from below the detection limit of the EDXRF up to 0.64%, while both cobalt (Co) and nickel (Ni) reach no more than 0.03%. Selenium (Se) and bismuth (Bi) are mostly below the detection limit, but are detectable in some samples up to 0.007 and 0.05% respectively. The trace-element patterns of the fragments from the other sites are similar to those of Santa Vittoria. However, nickel, silver and antimony of the *bronzetti* from Su Monte tend to have slightly higher concentrations (see Table 2; Fig 5).

Overall, all elemental values are in good agreement with the ICP-Q-MS results of the same samples (Table 3), which show deviations of maximum 15% for the main elements and no more than 20% for the minor and trace elements. Only the deviations of arsenic, bismuth, and antimony are higher, due to spectral interferences with lead and tin during XRF analysis. Given the differing analytical precision and sensitivity of the two methods as well as the differing sample quantities analysed, the relative differences observed are acceptable. Nevertheless, for arsenic, antimony, and bismuth, the values obtained by ICP-Q-MS are preferred, if available. This also applies to the indium and tellurium concentrations of the bronzes, which are only detectable by mass spectrometry in concentrations between 0.00005 and 0.0012% and 0.00038 and 0.015%, respectively (Table 3). In general, indium is rarely monitored in the analysis of bronzes, even though it turned

**Table 2. Results of analyses by EDXRF in mass%. Numbers under preservation correspond to the level of corrosion: 1 – no corrosion; 2 – partly corroded; 3 – entirely corroded (data: D Berger).**

| Lab. no. | Location | Preservation | Cu | Fe | Co | Ni | As | Se | Ag | Sn | Sb | Te | Pb | Bi |
|---|---|---|---|---|---|---|---|---|---|---|---|---|---|---|
| MA-202494 | Su Monte | 1 | 97 | <0.05 | <0.01 | 0.02 | <0.01 | <0.005 | 0.044 | 0.73 | 0.036 | <0.005 | 1.86 | <0.01 |
| MA-202495 | Su Monte | 1 | 99 | <0.05 | <0.01 | 0.02 | <0.005 | <0.005 | <0.002 | 0.004 | 0.008 | <0.005 | 0.70 | <0.005 |
| MA-202496 | Su Monte | 1 | 100 | <0.05 | <0.01 | 0.01 | 0.014 | <0.005 | 0.011 | 0.003 | 0.006 | <0.005 | 0.16 | <0.005 |
| MA-202498 | Su Monte | 3 | 85 | 0.35 | 0.01 | 0.04 | 1.27 | 0.007 | 0.023 | 13.6 | 0.042 | <0.005 | 0.12 | <0.005 |
| MA-202501 | Su Monte | 1 | 90 | <0.05 | 0.01 | 0.03 | 0.15 | <0.005 | 0.052 | 5.9 | 0.020 | <0.005 | 3.9 | 0.02 |
| MA-202502 | Su Monte | 1 | 94 | <0.05 | 0.01 | 0.02 | 0.47 | 0.006 | 0.092 | 4.7 | 0.030 | <0.005 | 0.63 | <0.005 |
| MA-202503 | Su Monte | 1 | 95 | <0.05 | 0.03 | 0.03 | 0.76 | 0.006 | 0.155 | 3.8 | 0.034 | <0.005 | 0.59 | <0.005 |
| MA-202504 | Su Monte | 1 | 82 | 0.52 | 0.03 | 0.07 | 0.35 | <0.005 | 0.038 | 5.0 | 0.288 | <0.005 | 11.3 | 0.03 |
| MA-202505 | Su Monte | 1 | | | | | | | | | | | Pb | |
| MA-202506 | Su Monte | 1 | 86 | 0.49 | 0.05 | 0.06 | 0.26 | 0.006 | 0.079 | 11.7 | 0.261 | <0.005 | 1.12 | 0.01 |
| MA-202507 | Su Monte | 1 | 90 | <0.05 | 0.01 | 0.05 | 0.193 | <0.005 | 0.058 | 8.8 | 0.119 | <0.005 | 0.43 | <0.005 |
| MA-202508 | Su Monte | 1 | 89 | <0.05 | <0.01 | 0.05 | 0.15 | <0.005 | 0.055 | 8.0 | 0.133 | <0.005 | 2.08 | 0.02 |
| MA-202499 | Abini | 1 | 90 | <0.05 | 0.02 | 0.03 | 0.35 | 0.005 | 0.071 | 8.7 | 0.029 | <0.005 | 0.30 | <0.005 |
| MA-202500 | Abini | 1 | 92 | 0.52 | 0.01 | 0.01 | <0.005 | <0.005 | 0.031 | 6.8 | 0.013 | <0.005 | 0.47 | <0.005 |
| MA-202509 | Abini | 2 | 89 | <0.05 | <0.01 | 0.05 | 0.285 | 0.006 | 0.125 | 9.5 | 0.75 | <0.005 | 0.20 | 0.034 |
| MA-202497 | Unknown | 1 | 89 | 0.06 | 0.02 | 0.03 | 0.31 | 0.005 | 0.021 | 9.7 | 0.024 | <0.005 | 0.62 | 0.017 |
| MA-202510 | Unknown | 3 | 90 | <0.05 | 0.03 | 0.03 | 0.52 | 0.006 | 0.074 | 7.5 | 0.020 | 0.009 | 1.51 | <0.01 |
| MA-202511 | Unknown | 2 | 90 | <0.05 | 0.01 | 0.02 | 0.22 | 0.009 | 0.017 | 9.3 | 0.016 | <0.005 | 0.88 | <0.005 |
| MA-202512 | Unknown | 3 | 81 | 0.21 | <0.01 | 0.01 | 0.10 | <0.005 | 0.035 | 17.5 | 0.034 | <0.005 | 1.25 | <0.01 |
| MA-202513 | Unknown | 2 | 89 | <0.05 | 0.02 | 0.04 | 0.43 | 0.007 | 0.029 | 9.9 | 0.041 | <0.005 | 0.40 | <0.005 |
| MA-202514 | Unknown | 2 | 94 | <0.05 | 0.03 | 0.04 | 0.48 | <0.005 | 0.050 | 4.8 | 0.098 | <0.005 | 0.69 | <0.005 |
| MA-202515 | Santa Vittoria | 1 | 88 | <0.05 | <0.01 | 0.03 | <0.01 | <0.005 | 0.055 | 8.7 | 0.039 | <0.005 | 2.73 | 0.07 |
| MA-202516 | Santa Vittoria | 1 | 93 | <0.05 | 0.01 | 0.02 | 0.64 | 0.008 | 0.099 | 6.1 | 0.033 | <0.005 | 0.40 | <0.005 |
| MA-202517 | Santa Vittoria | 1 | 89 | <0.05 | 0.01 | 0.03 | 0.25 | 0.007 | 0.025 | 9.6 | 0.053 | <0.005 | 0.64 | <0.005 |
| MA-202518 | Santa Vittoria | 1 | 89 | <0.05 | 0.02 | 0.03 | 0.13 | 0.006 | 0.037 | 8.0 | 0.075 | <0.005 | 2.33 | <0.01 |
| MA-202519 | Santa Vittoria | 1 | 89 | <0.05 | <0.01 | 0.02 | <0.01 | <0.005 | 0.057 | 7.8 | 0.039 | <0.005 | 2.78 | <0.01 |
| MA-202520 | Santa Vittoria | 1 | 91 | <0.05 | 0.02 | 0.04 | 0.30 | <0.005 | 0.037 | 7.8 | 0.110 | <0.005 | 1.08 | <0.01 |
| MA-202521 | Santa Vittoria | 1 | 89 | <0.05 | 0.02 | 0.03 | 0.28 | 0.009 | 0.017 | 8.7 | 0.037 | <0.005 | 2.11 | <0.01 |
| MA-202522 | Santa Vittoria | 1 | 89 | <0.05 | <0.01 | 0.02 | 0.064 | <0.005 | 0.056 | 10.4 | 0.088 | <0.005 | 0.45 | <0.005 |
| MA-202523 | Santa Vittoria | 1 | 93 | <0.05 | 0.03 | 0.02 | 0.62 | 0.007 | 0.130 | 6.0 | 0.026 | <0.005 | 0.53 | <0.005 |
| MA-202524 | Santa Vittoria | 1 | 90 | <0.05 | 0.03 | 0.03 | 0.41 | <0.005 | 0.024 | 9.2 | 0.045 | <0.005 | 0.60 | <0.005 |
| MA-202525 | Santa Vittoria | 1 | 88 | <0.05 | 0.01 | 0.03 | 0.22 | 0.006 | 0.021 | 10.1 | 0.036 | <0.005 | 1.37 | <0.01 |
| MA-202526 | Santa Vittoria | 1 | 91 | <0.05 | 0.02 | 0.03 | 0.31 | 0.005 | 0.048 | 8.4 | 0.052 | <0.005 | 0.17 | <0.005 |
| MA-202527 | Santa Vittoria | 1 | 86 | <0.05 | 0.02 | 0.03 | 0.18 | 0.007 | 0.012 | 12.3 | 0.021 | <0.005 | 1.66 | <0.01 |
| MA-202528 | Santa Vittoria | 1 | 89 | <0.05 | <0.01 | 0.02 | 0.09 | <0.005 | 0.043 | 9.7 | 0.074 | <0.005 | 1.33 | <0.01 |
| MA-202529 | Santa Vittoria | 1 | 88 | <0.05 | <0.01 | 0.03 | <0.01 | <0.005 | 0.062 | 6.0 | 0.053 | <0.005 | 5.6 | <0.01 |
| MA-202530 | Santa Vittoria | 1 | 86 | <0.05 | <0.01 | 0.02 | 0.17 | <0.005 | 0.093 | 11.3 | 0.127 | <0.005 | 2.07 | 0.01 |
| MA-202531 | Santa Vittoria | 2 | 90 | <0.05 | <0.01 | 0.02 | 0.094 | <0.005 | 0.033 | 9.9 | 0.075 | <0.005 | 0.28 | <0.005 |
| MA-202532 | Santa Vittoria | 1 | 90 | <0.05 | 0.03 | 0.03 | 0.40 | 0.008 | 0.023 | 8.9 | 0.029 | <0.005 | 0.59 | 0.007 |
| MA-202533 | Santa Vittoria | 1 | 81 | <0.05 | <0.01 | 0.02 | <0.01 | <0.005 | 0.058 | 13.5 | 0.062 | 0.010 | 5.2 | <0.01 |
| MA-202534 | Santa Vittoria | 1 | 92 | <0.05 | 0.02 | 0.03 | 0.44 | 0.010 | 0.036 | 7.2 | 0.015 | 0.009 | 0.07 | <0.005 |
| MA-202535 | Santa Vittoria | 2 | 87 | <0.05 | 0.01 | 0.02 | 0.050 | <0.005 | 0.033 | 12.8 | 0.024 | <0.005 | 0.35 | <0.005 |
| MA-202536 | Santa Vittoria | 1 | 86 | <0.05 | <0.01 | 0.03 | 0.02 | <0.005 | 0.072 | 13.5 | 0.092 | <0.005 | 0.63 | <0.005 |
| MA-202537 | Santa Vittoria | 1 | 88 | <0.05 | 0.02 | 0.02 | 0.31 | 0.006 | 0.036 | 11.6 | 0.038 | <0.005 | 0.22 | 0.020 |
| MA-202538 | Santa Vittoria | 1 | 85 | <0.05 | <0.01 | 0.02 | 0.16 | 0.007 | 0.039 | 14.0 | 0.065 | <0.005 | 0.82 | <0.005 |

*(Continued)*

**Table 2.** (Continued)

| Lab. no. | Location | Preservation | Cu | Fe | Co | Ni | As | Se | Ag | Sn | Sb | Te | Pb | Bi |
|---|---|---|---|---|---|---|---|---|---|---|---|---|---|---|
| MA-202539 | Santa Vittoria | 1 | 88 | <0.05 | 0.01 | 0.03 | 0.179 | 0.005 | 0.044 | 11.9 | 0.052 | <0.005 | 0.18 | 0.007 |
| MA-202540 | Santa Vittoria | 2 | 93 | <0.05 | <0.01 | 0.02 | 0.21 | <0.005 | 0.043 | 6.0 | 0.017 | <0.005 | 0.98 | <0.005 |
| MA-202541 | Santa Vittoria | 1 | 86 | <0.05 | 0.01 | 0.03 | 0.10 | <0.005 | 0.045 | 11.8 | 0.053 | <0.005 | 2.35 | <0.01 |
| MA-202542 | Santa Vittoria | 1 | 87 | <0.05 | 0.02 | 0.03 | 0.34 | 0.008 | 0.021 | 11.5 | 0.036 | <0.005 | 0.85 | <0.005 |
| MA-202543 | Santa Vittoria | 2 | 83 | <0.05 | <0.01 | 0.01 | 0.045 | <0.005 | 0.082 | 16.1 | 0.075 | <0.005 | 0.37 | <0.005 |
| MA-202544 | Santa Vittoria | 1 | 91 | <0.05 | 0.01 | 0.03 | 0.20 | 0.007 | 0.028 | 7.6 | 0.046 | <0.005 | 0.81 | 0.009 |
| MA-202545 | Santa Vittoria | 2 | 87 | <0.05 | <0.01 | 0.01 | 0.028 | <0.005 | 0.060 | 12.1 | 0.046 | <0.005 | 0.28 | <0.005 |

out to be a potentially diagnostic element in the provenance determination of copper-based metals, just as it is in the investigation of metallic tin [57–62]. Gold is an additional diagnostic element (over and above Co, Ni, Ag, and to a lesser extent As, Sb, Se and Te), but was not determined in any of the bronzes across the various locations.

The three ingots from the sanctuary of Su Monte are characterised by extremely low-impurity patterns. Determined by EDXRF, several elements are below the detection limit (Co, As, Se, Te, Bi), while others are present in low concentrations (Ni, Ag, Sb) comparable to the bronze figurines from the same site. The lead content varies from 0.16 to 1.9%, while tin is present in one ingot (MA-202494) at 1% and almost absent in the other two (Table 2). Analysed by ICP-Q-MS, this picture is again not changed significantly (Table 3), but the indium and tellurium contents are among the lowest in our sample.

Strikingly, several elements in the Santa Vittoria figurines are linearly correlated, e.g., arsenic and antimony, which show an abnormal negative correlation in the sense that high arsenic goes with low antimony and vice versa (Fig 5a). Comparable trends are observed for other element pairs and may be either negative or positive, but can be more or less clear, e.g., for lead and indium (Fig 5). From all observations, *bronzetto* MA-202523 can be excluded, as it distinctly deviates from the recognised correlation trends (see Fig 5). This may be true for additional bronzes (e.g., MA-202516, MA-202530) analysed only by EDXRF, but this is not always easy to discern due to the higher analytical uncertainties for some elements (As, Sb, Pb) or higher limits of detection for others (Co, Te, Bi) in this data. Also worth noting is the fact that many trace elements – particularly cobalt, nickel, arsenic, silver, indium, and tellurium – show correlations with the lead isotope ratios (Fig 6). Furthermore, there seems to be a hyperbolic correlation between the lead isotope ratios and lead concentration, which becomes linear for part of the samples when the reciprocal lead concentration is used (Fig 6f). Both observations are remarkable, as one would not expect such relationships between trace elements and lead isotopes in bronze objects *per se*. The respective isotopic and chemical fingerprint of ores is transferred to the metal during smelting, but since chemistry and isotopic characteristics in the same deposit or mine are independent geochemical parameters, it would be a great coincidence if correlations such as those observed were to occur when smelting the same or different batches of ore. This background helps in interpreting the present dataset, which will be explained in detail further below.

Due to the smaller number of samples, relationships between the artefacts from Su Monte, or the other two sites, as revealed by the trace elements of the Santa Vittoria bronzes are difficult to discern, provided the figurines of each site are in any way related regarding raw materials and manufacture. However, from the available data of the Su Monte objects (Fig 5), the impression emerges that hardly any element correlates with any other, except for three bronzes (MA-202501, −02, −03) displaying positive correlations for nickel, arsenic, silver, antimony, and tellurium. A similar picture is provided by the lead isotope ratios, which seem to show potential correlations only with nickel, bismuth, and lead (Fig 6).

### 3.2. The lead isotope ratios

Also revealing is the lead isotope composition itself. $^{206}Pb/^{204}Pb$ of the *bronzetti* from Santa Vittoria range from 17.908 to 18.374, $^{207}Pb/^{204}Pb$ from 15.599 to 15.649, while $^{208}Pb/^{204}Pb$ range from 37.981 to 38.315 (Fig 7a–b; Table 4). These

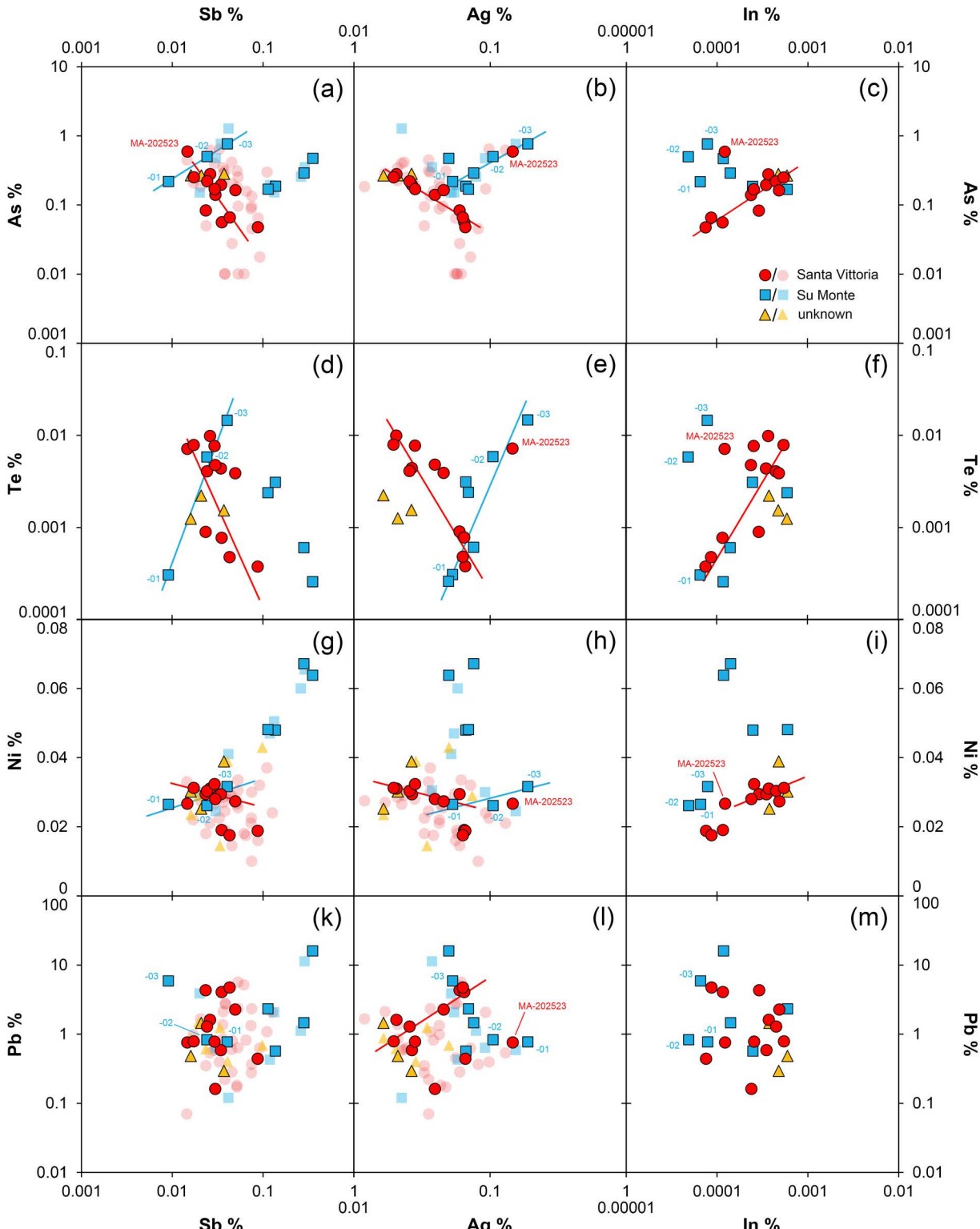

**Fig 5. Elemental scatter plots of the analysed *bronzetti* with data obtained through EDXRF (transparent symbols) and ICP-Q-MS (solid symbols).** The numbers correspond to the last two digits of the laboratory numbers listed in Table 1. Red and blue lines denote potential mixing lines. All values reported in mass% (diagrams: D Berger; data: D Berger, N Lockhoff).

**Table 3. Analytical results by ICP-Q-MS in mass% (Cu, Sn, Pb) and µg g⁻¹ (data: N Lockhoff). Only uncorroded samples have been analysed here.**

| Lab. no. | Location | Cu [%] | Mn | Fe | Co | Ni | Zn | As | Ag | In | Sn [%] | Sb | Te | Pb [%] | Bi |
|---|---|---|---|---|---|---|---|---|---|---|---|---|---|---|---|
| MA-202494 | Su Monte | 97 | 1.3 | 37 | 4.8 | 180 | <53 | 670 | 500 | <0.3 | 0.23 | 100 | 48 | 2.18 | 45 |
| MA-202495 | Su Monte | 99 | 7.6 | 31 | 19 | 220 | 46 | 170 | 1.8 | 0.5 | <0.0012 | 2.8 | <0.2 | 0.73 | 2.6 |
| MA-202496 | Su Monte | 100 | 1.8 | 37 | 19 | 79 | <16 | 170 | 100 | 0.2 | 0.0006 | 1.6 | <0.2 | 0.13 | 2.6 |
| MA-202501 | Su Monte | 88 | 1.6 | 400 | 100 | 260 | <43 | 2200 | 530 | 0.7 | 5.9 | 91 | 3.1 | 5.9 | 12 |
| MA-202502 | Su Monte | 94 | 0.5 | 210 | 110 | 260 | 72 | 5000 | 1050 | 0.5 | 4.8 | 240 | 58 | 0.83 | 17 |
| MA-202503 | Su Monte | 94 | 11.8 | 700 | 290 | 320 | 120 | 7600 | 1900 | 0.8 | 4.4 | 400 | 150 | 0.77 | 33 |
| MA-202504 | Su Monte | 77 | 0.9 | 4200 | 250 | 640 | 220 | 4700 | 500 | 1.2 | 5.7 | 3500 | <2.6 | 16.1 | 80 |
| MA-202506 | Su Monte | 86 | 4.0 | 5100 | 530 | 670 | 190 | 2900 | 760 | 1.4 | 11.4 | 2800 | 6.0 | 1.46 | 71 |
| MA-202507 | Su Monte | 89 | 9.3 | 110 | 93 | 480 | <75 | 1900 | 660 | 2.5 | 9.8 | 1400 | 31 | 0.57 | 60 |
| MA-202508 | Su Monte | 88 | 0.4 | 50 | 97 | 480 | 130 | 1700 | 700 | 6.0 | 9.5 | 1100 | 24 | 2.32 | 140 |
| MA-202500 | Abini | 92 | 2.0 | 4900 | 180 | 100 | 470 | 140 | 390 | 12 | 7.3 | 60 | <2.4 | 0.44 | 10 |
| MA-202497 | Unknown | 88 | 4.3 | 560 | 180 | 300 | 34 | 2700 | 210 | 5.9 | 10.7 | 160 | 13 | 0.49 | 130 |
| MA-202511 | Unknown | 89 | 4.9 | 360 | 100 | 250 | <51 | 2700 | 170 | 3.8 | 9.4 | 210 | 22 | 1.46 | 19 |
| MA-202513 | Unknown | 90 | 4.5 | 330 | 200 | 390 | 81 | 2800 | 270 | 4.8 | 9.6 | 370 | 15 | 0.29 | 31 |
| MA-202515 | Santa Vittoria | 86 | <0.2 | 70 | 22 | 290 | <54 | 830 | 600 | 2.9 | 9.4 | 230 | 9.0 | 4.3 | 650 |
| MA-202517 | Santa Vittoria | 90 | <0.4 | 95 | 130 | 290 | 79 | 1950 | 270 | 3.5 | 9.5 | 340 | 44 | 0.59 | 29 |
| MA-202519 | Santa Vittoria | 87 | 0.8 | 51 | 20 | 190 | 32 | 560 | 650 | 1.2 | 9.1 | 350 | 7.7 | 4.1 | 28 |
| MA-202521 | Santa Vittoria | 89 | 0.8 | 300 | 140 | 310 | 150 | 2800 | 210 | 3.7 | 9.0 | 260 | 99 | 1.61 | 46 |
| MA-202522 | Santa Vittoria | 89 | 1.9 | 57 | 16.6 | 190 | 98 | 480 | 660 | 0.8 | 10.1 | 870 | 3.8 | 0.44 | 37 |
| MA-202523 | Santa Vittoria | 92 | 2.5 | 81 | 230 | 270 | 130 | 5900 | 1500 | 1.2 | 6.4 | 150 | 72 | 0.76 | 23 |
| MA-202525 | Santa Vittoria | 88 | 2.4 | 180 | 130 | 300 | <100 | 2200 | 260 | 4.5 | 10.7 | 240 | 41 | 1.29 | 26 |
| MA-202533 | Santa Vittoria | 82 | 0.4 | 57 | 25 | 180 | <110 | 660 | 630 | 0.9 | 13.3 | 430 | <4.8 | 4.7 | 34 |
| MA-202539 | Santa Vittoria | 89 | 0.8 | 54 | 90 | 280 | 130 | 1400 | 390 | 2.4 | 10.3 | 300 | 48 | 0.16 | 42 |
| MA-202541 | Santa Vittoria | 86 | 2.4 | 65 | 120 | 270 | <120 | 1600 | 460 | 4.9 | 11.5 | 490 | 39 | 2.27 | 38 |
| MA-202542 | Santa Vittoria | 87 | 1.5 | 100 | 230 | 312 | <73 | 2500 | 200 | 5.5 | 12.1 | 170 | 79 | 0.78 | 23 |
| MA-202544 | Santa Vittoria | 91 | 0.3 | 110 | 100 | 320 | <77 | 1700 | 280 | 2.6 | 7.7 | 290 | 77 | 0.78 | 39 |

ranges are comparable to other Bronze Age copper and bronze artefacts from Sardinia, but the difference is the unusual trend observed in the uranogenic lead isotope ratios, where $^{207}Pb/^{204}Pb$ decreases with increasing $^{206}Pb/^{204}Pb$. This negative correlation cannot be explained by natural processes, since the relative abundances of the radiogenic fractions of $^{206}Pb$ and $^{207}Pb$ as well as $^{208}Pb$ all increase with time due to the radioactive decay of $^{238}U$, $^{235}U$ and $^{232}Th$ from the ore-forming process to the smelting event [63]. A negative trend in the lead isotope ratios implies an anthropogenic reason (see discussion section). Interestingly, the lead isotopic composition of the figurines with unknown provenance mirrors almost exactly that observed for most of the figurines from Santa Vittoria. Five fragments cluster around 17.93 $^{206}Pb/^{204}Pb$, 15.645 $^{207}Pb/^{204}Pb$ and 38.03 $^{208}Pb/^{204}Pb$, while a single sample has $^{206}Pb/^{204}Pb = 18.179$, $^{207}Pb/^{204}Pb = 15.616$ and $^{208}Pb/^{204}Pb = 38.319$ (Fig 7a–b; Table 4). Thus, these artefacts lie at both ends of the striking trend line defined for Santa Vittoria.

Two objects from Abini (MA-202499 and MA-202500) have also comparable isotope ratios, but the third one (MA-202509) is slightly different and shows ratios of $^{206}Pb/^{204}Pb = 18.227$, $^{207}Pb/^{204}Pb = 15.637$ and $^{208}Pb/^{204}Pb = 38.327$ respectively (Fig 7a–b; Table 4).

The lead isotope ratios of the Su Monte metal objects are distributed in a similar range as those of the Santa Vittoria bronzes, but unlike the latter, the uranogenic isotopes $^{206}Pb$ and $^{207}Pb$ display a positive trend (Fig 7a; Table 4). Nevertheless, there seems to be an intercept of both lines in the region between 17.98 and 18.00 ($^{206}Pb/^{204}Pb$) and around

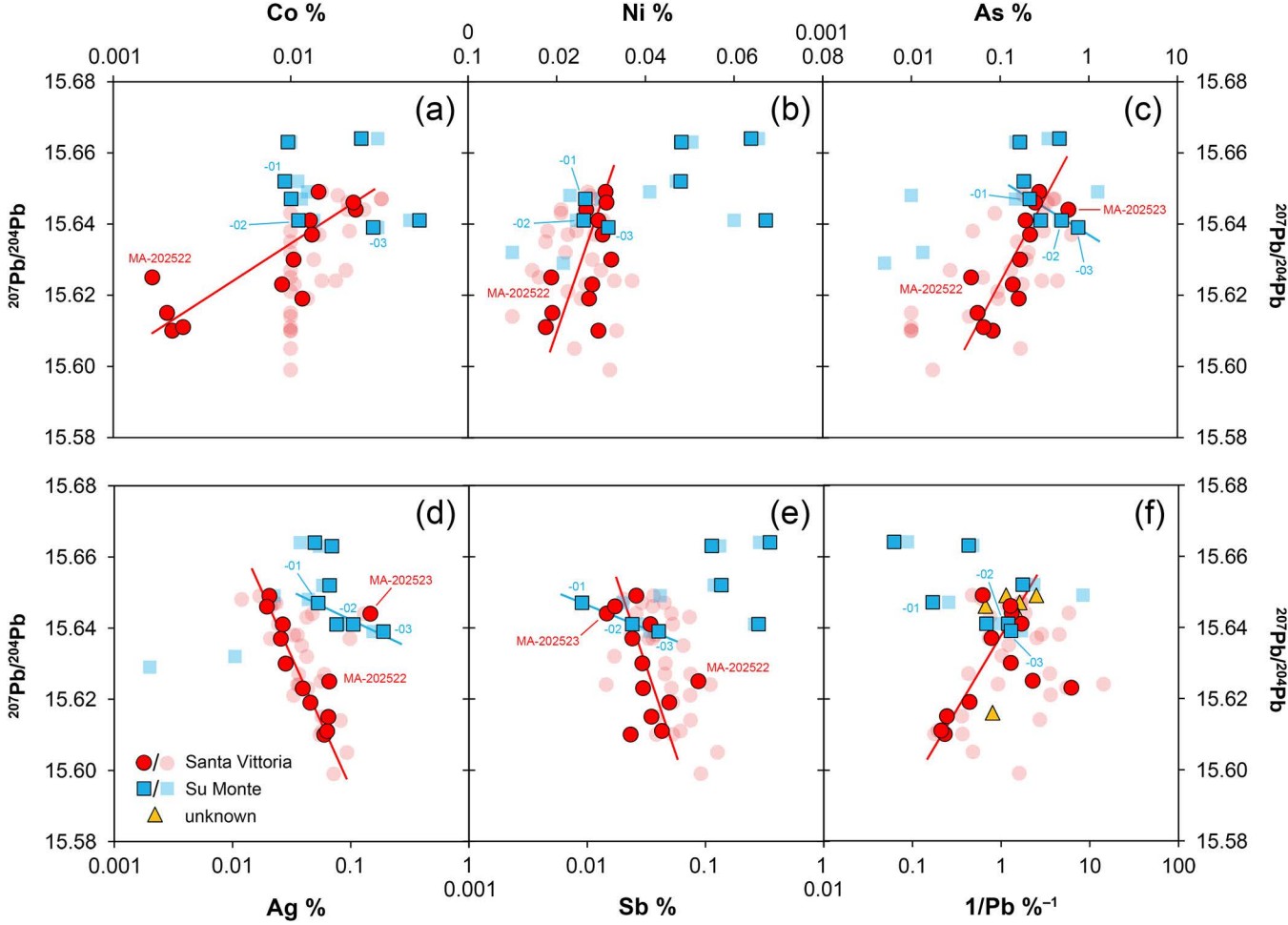

**Fig 6. Element concentrations by EDXRF (transparent symbols) and ICP-Q-MS (solid symbols) versus $^{207}Pb/^{204}Pb$ isotope ratios. Clear correlations of data (red lines) are observed for the figurines from Santa Vittoria and some from Su Monte. The numbers correspond to the last two digits of the laboratory numbers listed in Table 1. All concentrations in mass% (diagrams: D Berger; data: D Berger, B Höppner, N Lockhoff).**

15.63 ($^{207}Pb/^{204}Pb$), which is represented by two copper ingots (MA-202495 and −96) from Su Monte. The lead isotope composition of the third copper ingot (MA-202494) and of one statuette (MA-202501) deviates clearly from this line, and is in the same range as some figurines from Santa Vittoria and the unknown site (Fig 7a–b). Finally, the lead isotope composition of one lead sample from the foot of one figurine (MA-202505), which likely served to fix a figurine onto a stone or plinth in the sanctuary, is within the variation of the isotopic composition of the Su Monte bronzes (Fig 7a).

### 3.3. The tin and copper isotope composition

The stable isotope composition of copper and tin of the Santa Vittoria figurines shows a high overall variation (Fig 7c–d; Table 4; S2 Table). For tin, the isotope values δSn range from +0.044±0.001 to +0.096±0.002‰ u$^{-1}$ (Fig 7d), while for copper, $δ^{65}Cu$ is consistently negative, with values spreading from −1.27±0.01 to −0.18±0.01‰ (Fig 7c). However, a majority of eight bronzes display copper isotope values in a narrow range between −0.30±0.01 and −0.18±0.01‰. A combined view (Fig 8i) reveals small clusters of two and five bronzes characterised by nearly equal δSn and $δ^{65}Cu$. These clusters are variously reflected in the chemical and the tin isotope composition – especially in the concentrations of cobalt, nickel, arsenic, antimony, indium, and tellurium – but there are no correlative relationships (Fig 8). Clustering is even more

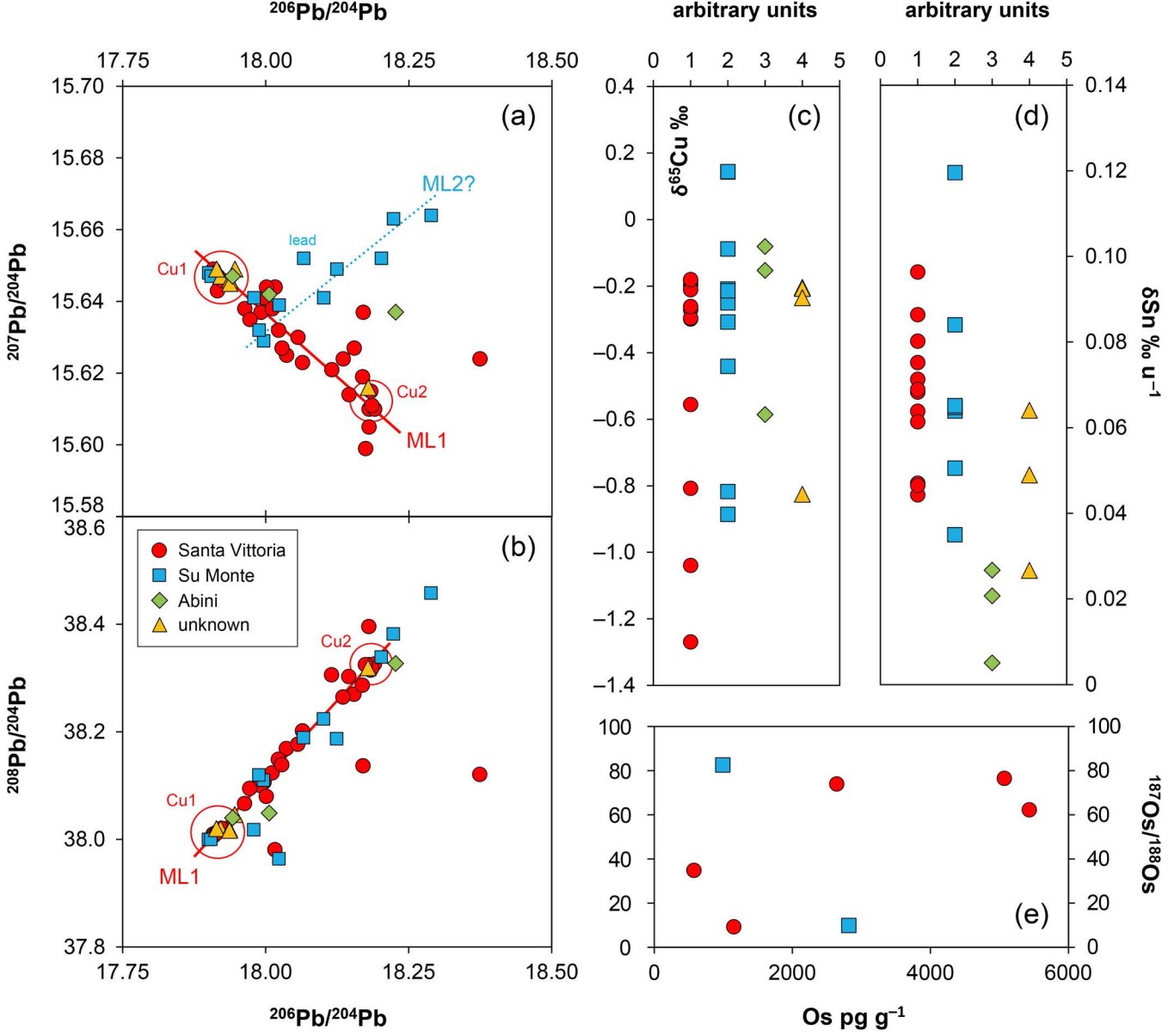

**Fig 7. Isotopic and chemical systematics of the analysed figurines.** (a–b) Lead isotope ratios, (c) copper isotope and (d) tin isotope values, (e) osmium isotope ratios and concentrations. Figs 7a and b show potential mixing lines (ML) with alleged copper pools Cu1 and Cu2 (diagrams: D Berger; data: M Brauns, G Brügmann, B Höppner).

pronounced for the copper isotope composition and the trace elements. A tight cluster is defined for those eight bronzes that are characterised by the small spread in the copper isotopy when plotted against the concentrations of cobalt, nickel, antimony, bismuth, indium, and tellurium. Clustering is less tight for arsenic, silver, and for tin and lead. Furthermore, there seems to be no correlation between copper isotopes and trace elements, nor between copper and lead isotopes. However, due to the non-proportional distribution of data points with a majority of closely distributed copper isotope values, a relationship cannot be ruled out in principle, as it may not be that evident from the small number of twelve samples.

**Table 4. Results of tin, copper, and lead isotope measurements. δSn and δSn$_{corr}$ (=δSn−0.025) in ‰ u$^{-1}$, δ$^{65}$Cu in ‰. For full tin isotope data cf. Supplementary Material S2 (data: G Brügmann, B Höppner).**

| Lab. no. | Location | δSn | δSn$_{corr}$ | 2SD | δ$^{65}$Cu | 2SD | $^{208}$Pb/$^{206}$Pb | 2SD | $^{207}$Pb/$^{206}$Pb | 2SD | $^{206}$Pb/$^{204}$Pb | 2SD | $^{208}$Pb/$^{204}$Pb | 2SD | $^{207}$Pb/$^{204}$Pb | 2SD |
|---|---|---|---|---|---|---|---|---|---|---|---|---|---|---|---|---|
| MA-202494 | Su Monte | | | | −0.21 | 0.06 | 2.1229 | 0.0001 | 0.87417 | 0.00001 | 17.900 | 0.001 | 38.000 | 0.003 | 15.648 | 0.001 |
| MA-202495 | Su Monte | | | | 0.14 | 0.02 | 2.1177 | 0.0001 | 0.86850 | 0.00002 | 17.996 | 0.001 | 38.110 | 0.003 | 15.629 | 0.001 |
| MA-202496 | Su Monte | | | | −0.82 | 0.004 | 2.1192 | 0.0001 | 0.86902 | 0.00002 | 17.988 | 0.001 | 38.120 | 0.002 | 15.632 | 0.001 |
| MA-202498 | Su Monte | | | | | | 2.1070 | 0.0001 | 0.86342 | 0.00002 | 18.124 | 0.002 | 38.187 | 0.005 | 15.649 | 0.001 |
| MA-202501 | Su Monte | 0.064 | 0.039 | 0.008 | −0.31 | 0.03 | 2.1224 | 0.0001 | 0.87391 | 0.00001 | 17.904 | 0.002 | 38.000 | 0.004 | 15.647 | 0.001 |
| MA-202502 | Su Monte | 0.120 | 0.095 | 0.001 | −0.44 | 0.02 | 2.1146 | 0.0001 | 0.86998 | 0.00002 | 17.979 | 0.001 | 38.018 | 0.003 | 15.641 | 0.002 |
| MA-202503 | Su Monte | 0.084 | 0.059 | 0.001 | −0.89 | 0.01 | 2.1064 | 0.0001 | 0.86771 | 0.00002 | 18.023 | 0.001 | 37.964 | 0.002 | 15.639 | 0.001 |
| MA-202504 | Su Monte | 0.065 | 0.040 | 0.001 | −0.25 | 0.01 | 2.1028 | 0.0001 | 0.85651 | 0.00001 | 18.289 | 0.001 | 38.458 | 0.003 | 15.664 | 0.001 |
| MA-202505 | Su Monte | | | | | | 2.1138 | 0.0001 | 0.86636 | 0.00002 | 18.066 | 0.001 | 38.189 | 0.001 | 15.652 | 0.001 |
| MA-202506 | Su Monte | 0.051 | 0.026 | 0.002 | −0.21 | 0.01 | 2.1118 | 0.0001 | 0.86410 | 0.00002 | 18.101 | 0.001 | 38.224 | 0.002 | 15.641 | 0.001 |
| MA-202507 | Su Monte | 0.065 | 0.040 | 0.001 | −0.09 | 0.02 | 2.1063 | 0.0001 | 0.85992 | 0.00002 | 18.202 | 0.002 | 38.339 | 0.006 | 15.652 | 0.002 |
| MA-202508 | Su Monte | 0.035 | 0.010 | 0.002 | 0.14 | 0.03 | 2.1063 | 0.0001 | 0.85954 | 0.00002 | 18.223 | 0.001 | 38.382 | 0.004 | 15.663 | 0.002 |
| MA-202499 | Abini | 0.021 | −0.004 | 0.006 | −0.59 | 0.004 | 2.1131 | 0.0001 | 0.86873 | 0.00002 | 18.006 | 0.002 | 38.049 | 0.004 | 15.642 | 0.001 |
| MA-202500 | Abini | 0.027 | 0.002 | 0.001 | −0.15 | 0.01 | 2.1202 | 0.0001 | 0.87206 | 0.00001 | 17.942 | 0.001 | 38.040 | 0.003 | 15.647 | 0.001 |
| MA-202509 | Abini | 0.005 | −0.020 | 0.004 | −0.08 | 0.01 | 2.1028 | 0.0001 | 0.85789 | 0.00002 | 18.227 | 0.002 | 38.327 | 0.002 | 15.637 | 0.001 |
| MA-202497 | Unknown | 0.064 | 0.039 | 0.003 | −0.20 | 0.001 | 2.1215 | 0.0001 | 0.87312 | 0.00002 | 17.921 | 0.001 | 38.020 | 0.001 | 15.647 | 0.001 |
| MA-202510 | Unknown | | | | | | 2.1197 | 0.0001 | 0.87229 | 0.00002 | 17.937 | 0.001 | 38.019 | 0.001 | 15.646 | 0.001 |
| MA-202511 | Unknown | 0.027 | 0.002 | 0.008 | −0.21 | 0.01 | 2.1223 | 0.0001 | 0.87352 | 0.00001 | 17.914 | 0.002 | 38.020 | 0.005 | 15.649 | 0.002 |
| MA-202512 | Unknown | | | | | | 2.1079 | 0.0001 | 0.85906 | 0.00002 | 18.179 | 0.001 | 38.319 | 0.001 | 15.616 | 0.001 |
| MA-202513 | Unknown | 0.049 | 0.024 | 0.005 | −0.23 | 0.01 | 2.1201 | 0.0001 | 0.87203 | 0.00001 | 17.946 | 0.001 | 38.046 | 0.001 | 15.649 | 0.001 |
| MA-202514 | Unknown | | | | −0.83 | 0.02 | 2.1195 | 0.0001 | 0.87226 | 0.00001 | 17.937 | 0.002 | 38.017 | 0.004 | 15.645 | 0.002 |
| MA-202515 | S. Vittoria | 0.096 | 0.071 | 0.002 | −0.20 | 0.01 | 2.1081 | 0.0001 | 0.85866 | 0.00002 | 18.180 | 0.001 | 38.325 | 0.001 | 15.610 | 0.001 |
| MA-202516 | S. Vittoria | | | | | | 2.0990 | 0.0001 | 0.86063 | 0.00003 | 18.170 | 0.001 | 38.137 | 0.003 | 15.637 | 0.001 |
| MA-202517 | S. Vittoria | 0.047 | 0.022 | 0.001 | −0.30 | 0.002 | 2.1175 | 0.0001 | 0.86909 | 0.00003 | 17.997 | 0.001 | 38.107 | 0.002 | 15.641 | 0.001 |
| MA-202518 | S. Vittoria | | | | | | 2.1081 | 0.0001 | 0.86081 | 0.00001 | 18.154 | 0.001 | 38.270 | 0.001 | 15.627 | 0.001 |
| MA-202519 | S. Vittoria | 0.080 | 0.055 | 0.001 | −1.27 | 0.01 | 2.1072 | 0.0001 | 0.85879 | 0.00001 | 18.183 | 0.001 | 38.315 | 0.003 | 15.615 | 0.001 |
| MA-202520 | S. Vittoria | | | | | | 2.1100 | 0.0001 | 0.86154 | 0.00001 | 18.135 | 0.001 | 38.265 | 0.002 | 15.624 | 0.001 |
| MA-202521 | S. Vittoria | 0.064 | 0.039 | 0.002 | −0.27 | 0.01 | 2.1226 | 0.0001 | 0.87389 | 0.00002 | 17.908 | 0.002 | 38.010 | 0.004 | 15.649 | 0.002 |
| MA-202522 | S. Vittoria | 0.044 | 0.019 | 0.001 | −1.04 | 0.03 | 2.1163 | 0.0001 | 0.86634 | 0.00002 | 18.036 | 0.002 | 38.169 | 0.005 | 15.625 | 0.002 |
| MA-202523 | S. Vittoria | 0.061 | 0.036 | 0.001 | −0.56 | 0.02 | 2.1082 | 0.0001 | 0.86836 | 0.00002 | 18.016 | 0.001 | 37.981 | 0.002 | 15.644 | 0.001 |
| MA-202524 | S. Vittoria | | | | | | 2.1219 | 0.0001 | 0.87344 | 0.00001 | 17.914 | 0.002 | 38.012 | 0.003 | 15.647 | 0.001 |
| MA-202525 | S. Vittoria | 0.075 | 0.050 | 0.002 | −0.30 | 0.01 | 2.1178 | 0.0001 | 0.86918 | 0.00001 | 17.991 | 0.002 | 38.100 | 0.006 | 15.637 | 0.002 |
| MA-202526 | S. Vittoria | | | | | | 2.1154 | 0.0001 | 0.86903 | 0.00003 | 18.001 | 0.002 | 38.080 | 0.004 | 15.644 | 0.002 |
| MA-202527 | S. Vittoria | | | | | | 2.1222 | 0.0001 | 0.87376 | 0.00001 | 17.909 | 0.001 | 38.007 | 0.003 | 15.648 | 0.001 |
| MA-202528 | S. Vittoria | | | | | | 2.1219 | 0.0001 | 0.87316 | 0.00001 | 17.915 | 0.002 | 38.014 | 0.005 | 15.643 | 0.002 |
| MA-202529 | S. Vittoria | | | | | | 2.1070 | 0.0001 | 0.85819 | 0.00003 | 18.190 | 0.001 | 38.326 | 0.004 | 15.610 | 0.002 |
| MA-202530 | S. Vittoria | | | | | | 2.1120 | 0.0001 | 0.85836 | 0.00001 | 18.180 | 0.001 | 38.396 | 0.002 | 15.605 | 0.001 |
| MA-202531 | S. Vittoria | | | | | | 2.1146 | 0.0001 | 0.86232 | 0.00002 | 18.115 | 0.001 | 38.306 | 0.002 | 15.621 | 0.001 |
| MA-202532 | S. Vittoria | | | | | | 2.1218 | 0.0001 | 0.87337 | 0.00001 | 17.916 | 0.001 | 38.013 | 0.001 | 15.647 | 0.001 |
| MA-202533 | S. Vittoria | 0.086 | 0.061 | 0.002 | −0.81 | 0.01 | 2.1071 | 0.0001 | 0.85844 | 0.00001 | 18.185 | 0.002 | 38.318 | 0.003 | 15.611 | 0.001 |

(Continued)

**Table 4.** (Continued)

| Lab. no. | Location | δSn | 2SD | δSn$_{corr}$ | 2SD | δ$^{65}$Cu | 2SD | $^{208}$Pb/$^{206}$Pb | 2SD | $^{207}$Pb/$^{206}$Pb | 2SD | $^{206}$Pb/$^{204}$Pb | 2SD | $^{208}$Pb/$^{204}$Pb | 2SD | $^{207}$Pb/$^{204}$Pb | 2SD |
|---|---|---|---|---|---|---|---|---|---|---|---|---|---|---|---|---|---|
| MA-202534 | S. Vittoria | | | | | | | 2.0748 | 0.0001 | 0.85034 | 0.00001 | 18.374 | 0.001 | 38.121 | 0.003 | 15.624 | 0.001 |
| MA-202535 | S. Vittoria | | | | | | | 2.1192 | 0.0001 | 0.87055 | 0.00001 | 17.963 | 0.002 | 38.067 | 0.002 | 15.638 | 0.001 |
| MA-202536 | S. Vittoria | | | | | | | 2.1089 | 0.0001 | 0.85835 | 0.00001 | 18.174 | 0.001 | 38.325 | 0.002 | 15.599 | 0.001 |
| MA-202537 | S. Vittoria | | | | | | | 2.1167 | 0.0001 | 0.86825 | 0.00001 | 18.011 | 0.002 | 38.124 | 0.005 | 15.638 | 0.002 |
| MA-202538 | S. Vittoria | | | | | | | 2.1197 | 0.0001 | 0.86998 | 0.00001 | 17.972 | 0.001 | 38.095 | 0.002 | 15.635 | 0.002 |
| MA-202539 | S. Vittoria | 0.047 | | 0.022 | 0.002 | −0.26 | 0.04 | 2.1148 | 0.0001 | 0.86488 | 0.00001 | 18.064 | 0.002 | 38.202 | 0.004 | 15.623 | 0.001 |
| MA-202540 | S. Vittoria | | | | | | | 2.1167 | 0.0001 | 0.86736 | 0.00002 | 18.022 | 0.001 | 38.149 | 0.001 | 15.632 | 0.001 |
| MA-202541 | S. Vittoria | 0.071 | | 0.046 | 0.002 | −0.18 | 0.004 | 2.1073 | 0.0001 | 0.85963 | 0.00002 | 18.169 | 0.001 | 38.287 | 0.002 | 15.619 | 0.001 |
| MA-202542 | S. Vittoria | 0.068 | | 0.043 | 0.002 | −0.21 | 0.01 | 2.1213 | 0.0001 | 0.87292 | 0.00002 | 17.923 | 0.003 | 38.021 | 0.005 | 15.646 | 0.002 |
| MA-202543 | S. Vittoria | | | | | | | 2.1110 | 0.0001 | 0.86051 | 0.00002 | 18.145 | 0.001 | 38.303 | 0.002 | 15.614 | 0.001 |
| MA-202544 | S. Vittoria | 0.069 | | 0.044 | 0.001 | −0.18 | 0.01 | 2.1144 | 0.0001 | 0.86567 | 0.00002 | 18.056 | 0.001 | 38.177 | 0.002 | 15.630 | 0.001 |
| MA-202545 | S. Vittoria | | | | | | | 2.1155 | 0.0001 | 0.86683 | 0.00003 | 18.028 | 0.003 | 38.139 | 0.008 | 15.627 | 0.003 |

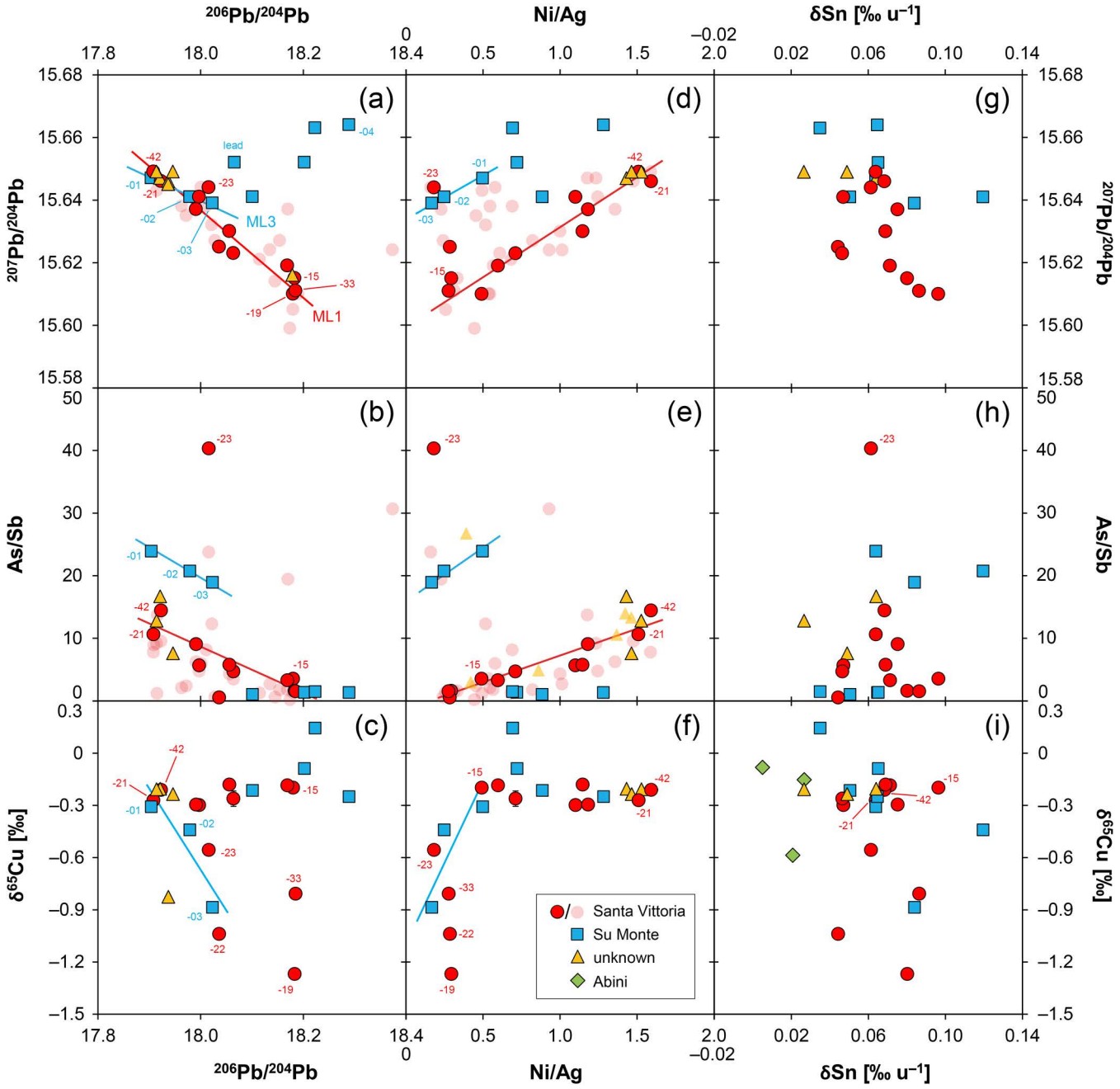

**Fig 8. Scatter plots of isotopic and elemental ratios of the studied artefacts.** Transparent symbols correspond to samples with chemical data by EDXRF, while solid symbols indicate data by ICP-Q-MS. The numbers correspond to the last two digits of the laboratory numbers listed in Table 1. Red and blue lines denote potential mixing lines (ML) (diagrams: D Berger; data: D Berger, G Brügmann, B Höppner, N Lockhoff).

The stable isotope compositions for the Su Monte figurines are as variable as the compositions of the bronzes from the other sanctuary (Fig 7c–d; Table 4). δSn is again consistently positive and ranges from +0.035±0.002 to +0.120±0.001‰ u⁻¹, thus only slightly exceeding the range documented for Santa Vittoria. The variability of $\delta^{65}$Cu is also comparable, with an ingot (MA-202494) and three figurines falling slightly below 0, between −0.44±0.02 and −0.21±0.01‰ (Fig 7C). Another

ingot (MA-202496) and a figurine (MA-202503) are characterised by values around –0.85‰, while the third ingot (MA-202495) and one *bronzetto* (MA-202508) are slightly positive at 0.14‰. This makes the latter the only two objects in the entire sample series to show copper isotope values above 0. The small dataset does again not allow to clearly determine any correlations of stable isotope ratios with each other and of stable isotopes with trace elements and lead isotopes.

The δSn of the figurines from Abini, on the other hand, are the lowest of all the bronzes analysed. They have δSn values from +0.005±0.004 to +0.027±0.001‰ u⁻¹, whereas the three bronzes from the unknown site (MA-202497, −511, −513) are similar to the objects from Su Monte and Santa Vittoria, with +0.027±0.008 to +0.064±0.003‰ u⁻¹. The latter is also true for their copper isotope composition, which is identical to the group of eight samples from Santa Vittoria, with δ⁶⁵Cu between –0.23±0.01 and –0.20±0.001‰ (Fig 7C–d; Table 4). A fourth statuette from the unknown location is isotopically very negative at –0.83±0.02‰ and thus differs from the other objects. The variability of δ⁶⁵Cu of the Abini bronzes for their part is within the range of the other finds (Fig 7C).

### 3.4. The osmium isotope ratios and concentrations

The determination of the osmium isotopic composition ¹⁸⁷Os/¹⁸⁸Os and the osmium concentration is part of an unprecedented pilot study on copper-based artefacts that aims to investigate the significance of osmium isotopes in reconstructing the origin of metals. This study is also expected to provide insights into technological processes. Due to the tendency of osmium to form volatile oxides under oxidising conditions, significant osmium losses are anticipated during the roasting of copper ores, resulting in very low osmium contents in the corresponding metal. In contrast, for oxidic copper ores that do not require roasting, no osmium loss is expected, which should result in significantly higher concentrations in the smelted copper.

Five samples from Santa Vittoria were selected for this purpose according to their ²⁰⁶Pb/²⁰⁴Pb and ²⁰⁷Pb/²⁰⁴Pb ratios, so that they were evenly distributed along the trend line defined by these ratios (Fig 9; Table 1). Two additional samples from an ingot (MA-202494) and a *bronzetto* from Su Monte were randomly chosen, while no *bronzetti* of Abini and the unknown site have been measured at this state of research.

¹⁸⁷Os/¹⁸⁸Os proved very high for all statuettes with values from 9.3055±0.0016 to 76.5588±0.0319, while the ingot from Su Monte even reaches 82.6111±0.0003 (Table 5; Fig 7e). Conspicuously, the ingot and one of the figurines from Santa Vittoria (MA-202521) with a similarly high osmium isotope ratio have matching lead isotope values (Table 3).

The osmium concentration of all objects is very high, but also highly variable within one order of magnitude between 580–5400 pg g⁻¹ (Fig 7e). There could be a correlation between the osmium isotope composition and the osmium concentration in the metals (Fig 9b), though this is not easy to establish based on the small sample set. The unclear trend could be reproduced in correlations between the osmium and the lead isotope composition (Fig 9b) as well as some trace elements (not shown).

## 4. Discussion

### 4.1. Comparison with other Bronze Age metalwork from Sardinia

Numerous chemical analyses of Sardinian metal objects have been carried out in past decades. These analyses included about fifty anthropomorphic and zoomorphic figurines and about twenty-five miniature boats, as well as weapons, tools, jewellery, and other object types [6,11–15,43,45,46,64,65]. In addition, chemical data from hundreds of copper ingots is available today [6,10,15,47,64]. Analytical methods were quite diverse across decades, ranging from surface analysis with portable XRF, stationary XRF, and neutron activation analysis on drill samples, to atomic absorption/emission spectroscopy and inductively coupled plasma ionisation mass spectrometry on sample solutions. Due to this diversity in methods, and to differences in analytical sensitivity and precision, the datasets are not directly comparable. Nevertheless, the data can be used for generic comparison with our recent dataset, e.g., in the form of histograms, and for a basic discussion.

Histograms for key element concentrations are shown in Fig 10. Generally, the figurines of the present paper show a similar elemental range and distribution to those figurines analysed in earlier studies. Only the frequency of individual

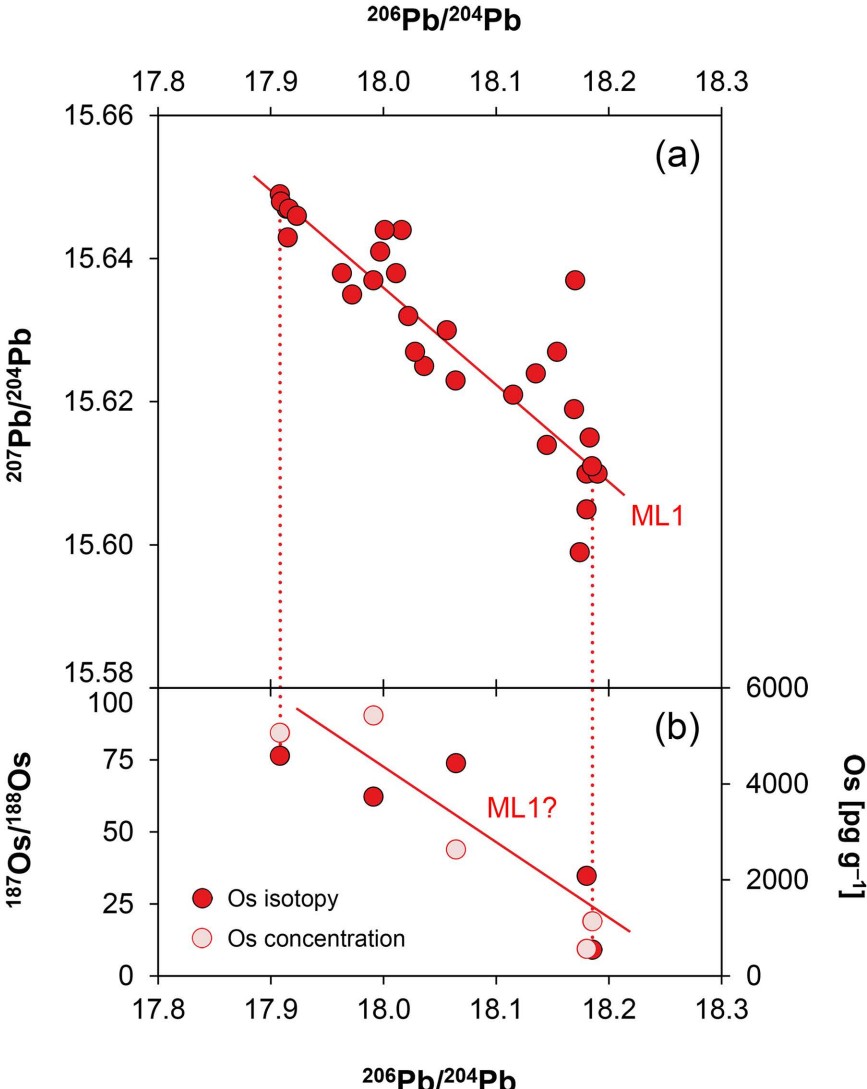

**Fig 9. Lead isotope versus osmium isotope ratios and osmium concentrations of the Santa Vittoria *bronzetti* with potential mixing line (diagrams: D Berger; data: M Brauns, B Höppner).**

**Table 5. Osmium isotope composition and osmium concentration (in pg g⁻¹) of the analysed samples (data: M Brauns).**

| Lab. no. | Location | Object | $^{187}Os/^{188}Os$ | 2SD | Os | 2SD |
|---|---|---|---|---|---|---|
| MA-202494 | Su Monte | Copper ingot | 82.6111 | 0.0003 | 993 | 9 |
| MA-202506 | Su Monte | *Bronzetto* | 9.9483 | 0.0301 | 2817 | 26 |
| MA-202515 | Santa Vittoria | *Bronzetto* | 34.9171 | 0.0064 | 575 | 5 |
| MA-202521 | Santa Vittoria | *Bronzetto* | 76.5588 | 0.0319 | 5070 | 47 |
| MA-202525 | Santa Vittoria | *Bronzetto* | 62.3468 | 0.0172 | 5433 | 50 |
| MA-202533 | Santa Vittoria | *Bronzetto* | 9.3055 | 0.0016 | 1152 | 11 |
| MA-202539 | Santa Vittoria | *Bronzetto* | 74.0326 | 0.0257 | 2642 | 25 |

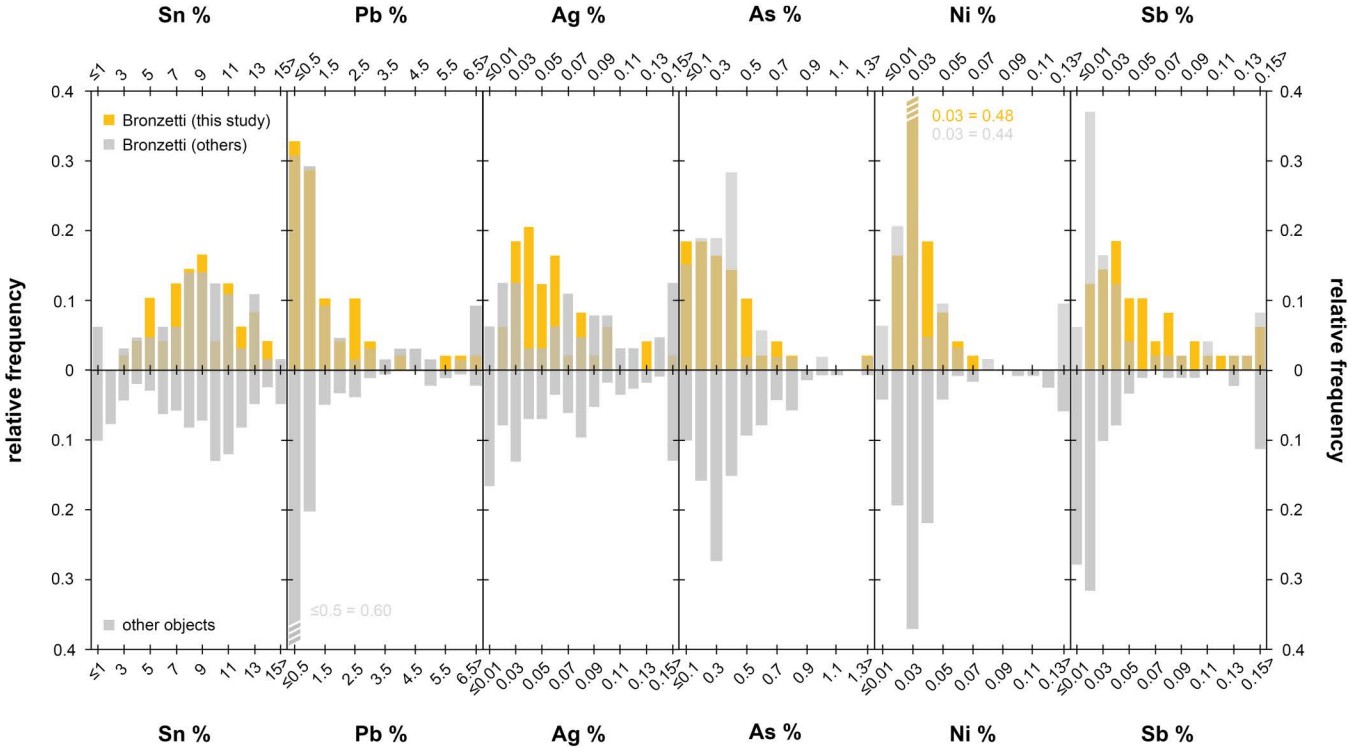

**Fig 10. Elemental distribution histograms of the *bronzetti* of the present study compared with figurines and other Sardinian metal artefacts of the Late Bronze and Early Iron Age analysed in previous studies (diagrams: D Berger; comparative data: references given in the text).**

concentrations tends to vary, likely caused by the different analytical methods used. However, although only a limited number of trace elements (Ni, As, Ag, Sb, Pb) are available for comparison, this number of elements is sufficient at first instance, as they are all diagnostic for the ore base and its origin [56,66]. According to the observed distribution patterns, one would thus infer similar ore bases for most figurines and probably the same origin of ores for a portion of them. This conclusion might also be true for other contemporary Sardinian metalwork (Fig 10). The data is, again, comparable with that of the figurines; however, in part, there are slight differences, most likely because the dataset of comparative bronzes also contains imported items, objects that may reflect other ores of different origin. Yet, as already stressed by Balmuth and Tylecote [10], conclusions about metal provenance based on chemical composition alone are problematic, or not possible, as this is not an unambiguous parameter. Lead isotope and likely copper isotope compositions must be included and assessed together with the chemical signature to obtain statements on a broader base. Nonetheless, the chemical patterns both of the figurines and of the other bronzework of the Sardinian Recent/Final Bronze Age reveal a uniform picture.

### 4.2. Source mixing and provenance of raw materials

Previous sections have put forward the hypothesis of source mixing during the manufacture of the studied bronze objects. This hypothesis is most convincingly confirmed for the *bronzetti* from Santa Vittoria and Su Monte, due to the larger number of samples from these two sites. For this reason, we will concentrate on these two localities in what follows.

The mixing and recycling of metals have been discussed since the birth of archaeometallurgical research. Discussion has gained momentum in recent years due to large-scale research projects and programmes. New approaches were tested on Bronze Age artefacts to see whether scientific data can distinguish mixing practices. It turned out that the most promising proxies for mixing are the lead isotope composition and its combination with other isotope systems and with the

chemical composition, in addition to archaeological and typological measures [e.g., 61,67–70]. In the present setting, the great value of lead isotopy is worth stressing, as especially in the 1990s this analytical method was deemed of low value [e.g., 20]. In fact, if we assume multiple/continued mixing and recycling of materials or ore sources in the way Pollard and co-workers suggested it for arsenical copper [71–73], it would be almost impossible to recognise mixing, because chemical and isotopic signatures would change drastically and supposedly equilibrate. However, the question must be posed: Was metal really mixed or recycled repeatedly, or are we more likely dealing with a lesser degree of mixing? Did mixing and recycling take place at all? The latter scenario is rarely challenged any more, but there is still disagreement about the extent of these metallurgical practices. The *bronzetti* may help to disentangle the problem area, even if only partially.

**4.2.1. Cypriot copper for the bronzetti?.** Although there are chronological differences between the figurines (c. 1000–800 BCE) and the peak production and distribution of oxhide ingots (c. 1300–1100 BCE), it is important to briefly address the potential use of Cypriot copper in the production of the *bronzetti*. There has been a lively debate in past decades about why Sardinia imported copper from Cyprus in ingot form, yet seemingly did not use it for the manufacture of local products [1,6,20,22,74,75]. Without delving into detailed arguments here, there is a consensus (except for [67]) that the Nuragic metallurgists did not use copper from Cyprus for their local products. The current dataset supports this commonly held belief, as the lead isotope ratios of the bronzes can by no means be reconciled with those of Cypriot copper ores and oxhide ingots (Fig 11), even when mixing of lead-poor copper from Cyprus with lead or lead-rich copper from elsewhere is considered (detailed discussion on the possibility of mixing is given in S3 Appendix). The same conclusion is to be inferred from the chemical composition of the figurines, which generally have higher concentrations of silver, antimony, and lead (Fig 12). Furthermore, no correlation trends are observed in any elemental combinations between the Nuragic items and the oxhide ingots, as would be expected if two copper sources were mixed. In conclusion, Cypriot copper stands out as a highly unlikely source for these local Nuragic products.

**4.2.2. Santa Vittoria. 4.2.2.1. Analytical evidence for source mixing:**
The lead isotope ratios of the Santa Vittoria *bronzetti* fragments provide the clearest indication of source mixing. Contrary to natural principles, the uranogenic isotopes $^{206}$Pb and $^{207}$Pb (from radioactive decay $^{238}$U and $^{235}$U) show a distinct negative correlation when normalised to $^{204}$Pb (Fig 7a). Only a few objects deviate from this trend (MA-20216,-23,-30,-34 and-36).

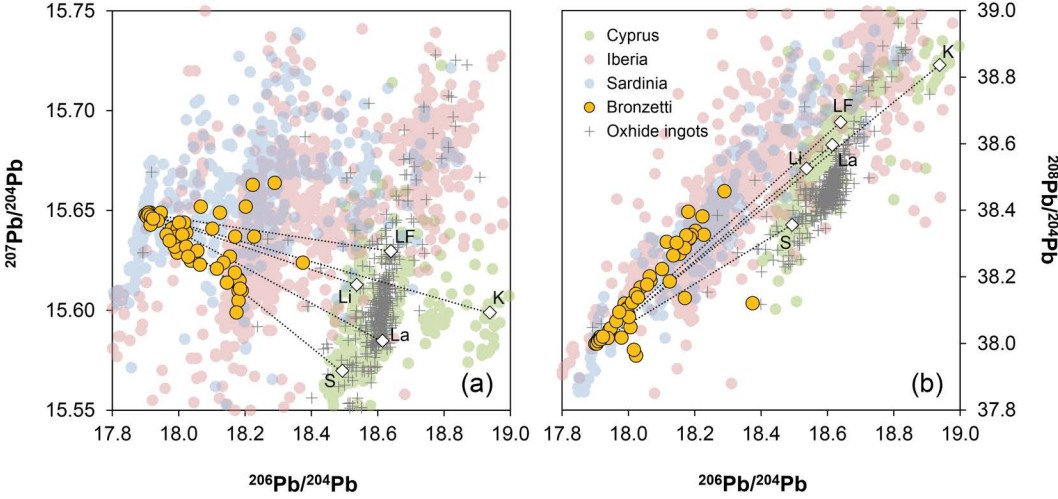

**Fig 11. Lead isotope ratios of the studied figurines compared with those of copper and lead ores from Sardinia, Iberia, and Cyprus.** The dotted lines represent mixing lines, which are expected if there was mixing between copper from Cu1 copper pool and Cypriot ores from the Solea (S), Larnaca (La), Limni (Li), Limmasol Forest (LF), and Kalavassos (K) axes. The data points of Cypriot deposits (white diamonds) correspond to the mean isotope ratios for all ores within the respective regions (diagrams: D Berger; data from ingots and ores from references given in S4 Appendix).

The same deviation is also evident in the $^{206}Pb/^{204}Pb$–$^{208}Pb/^{204}Pb$ diagram that includes the thorogenic isotope $^{208}Pb$ (from $^{232}Th$ radioactive decay), while the other bronzes are still well aligned (Fig 7b). However, the observed positive trend is indistinguishable from a natural one. Since the $^{208}Pb$, $^{206}Pb$ and $^{207}Pb$ isotopes have different origins (see above), dissimilar trends are no argument against mixing. For example, one component might have lower $^{207}Pb/^{204}Pb$ ratios than the other, while still showing higher $^{208}Pb/^{204}Pb$ ratios. In such a scenario, opposing data trends in combination with $^{206}Pb/^{204}Pb$ would emerge. Thus, regardless of the different orientations, the concurrent alignment of the data points across both diagrams is a robust evidence supporting the hypothesis advanced, namely that most of the Santa Vittoria metal objects must be a result of mixing. In the same diagrams, a clustering of data points around $^{206}Pb/^{204}Pb \approx 18.185$, $^{207}Pb/^{204}Pb \approx 15.61$, and $^{208}Pb/^{204}Pb \approx 38.32$ is observed, which is consistent with the highest lead concentrations in three statuettes (MA-202515, −29 and −33). Another clustering is evident for $^{206}Pb/^{204}Pb \approx 17.915$, $^{207}Pb/^{204}Pb \approx 15.645$, $^{208}Pb/^{204}Pb \approx 38.0$ (Fig 7a–b).

In addition to the lead isotope ratios, correlations are observed for trace-element concentrations, and even for lead isotopes and trace elements (Figs 5–6). Relying on the ICP-Q-MS data, the most obvious correlations are found for the element

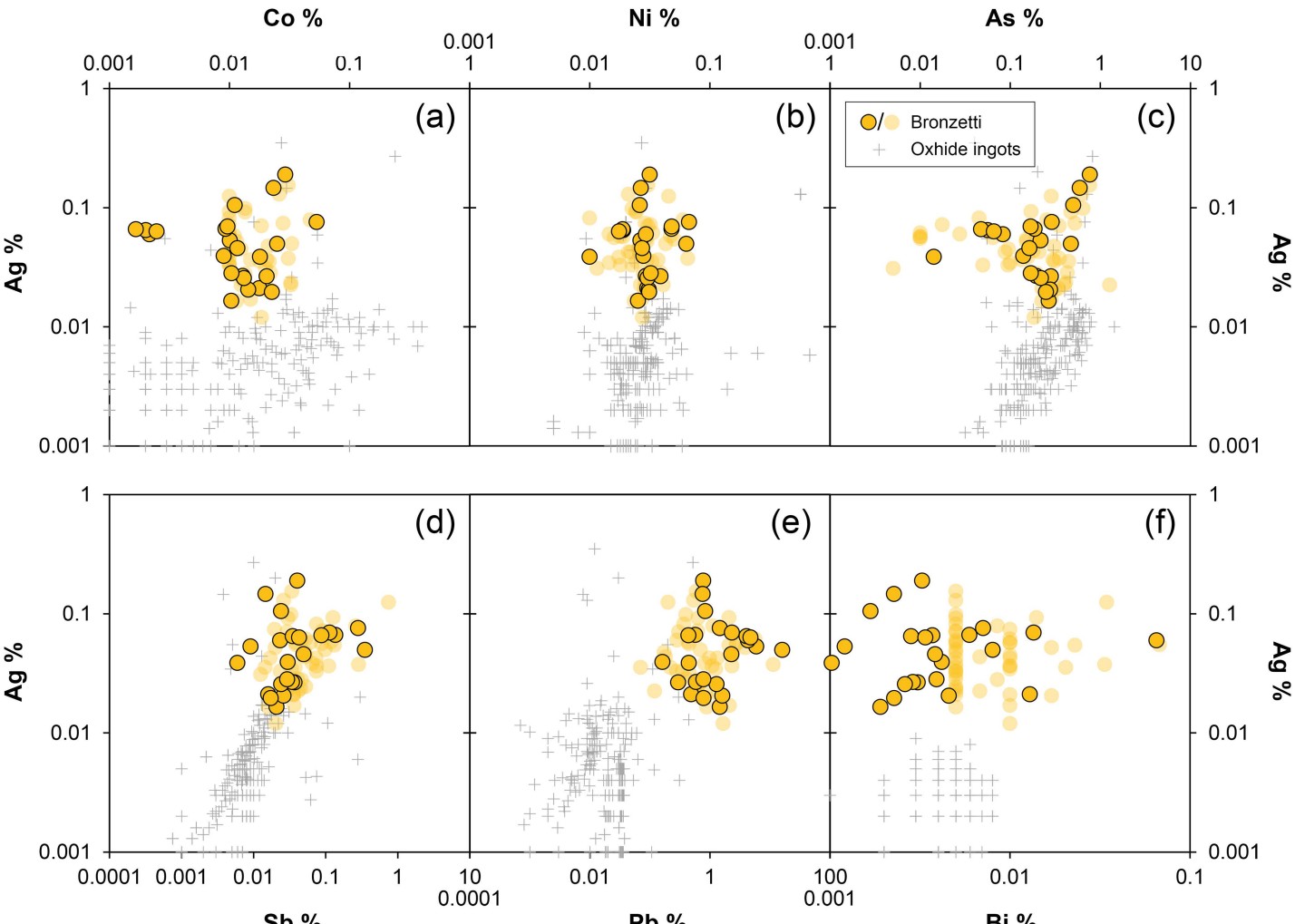

**Fig 12. Elemental composition of the analysed figurines compared with that of oxhide and bun ingots of Cypriot copper.** All concentrations in mass% (diagrams: D Berger; data from ingots and ores from references given in S4 Appendix).

pairs Ag/In (negative), Ag/Te (negative), Ag/As (negative), As/Sb (negative), Ni/Ag (slightly negative), Ni/In (positive) and As/In (positive), while other pairs show less clear or no relationships at all. According to Pernicka [56], these are the elements that are indicative of metal origin and processing; that is, they are only slightly affected during pyrometallurgical processing steps. As mentioned above, negative trends are a strong indication of mixing. For example, negative correlation between arsenic and antimony implies that a material rich in arsenic and poor in antimony has been mixed with another source poor in arsenic and rich in antimony (in different proportions). If no relationship existed between the examined artefacts consisting of fragments of arms and feet, respectively, this behaviour is not to be expected. Either a positive correlation should emerge, due to the association of both elements in many copper deposits, or a data cloud without any systematics. Since the same considerations apply to other element combinations with negative as well as positive correlations, and since element ratios (e.g., As/Sb vs. Ni/Ag) correlate well (Fig 8e), one must infer from this finding the mixing of two materials with different chemical compositions.

This interpretation fits well with correlations in element concentrations or element ratios and the lead isotope composition. Significant relationships are found, e.g., for Ni/Ag, In/Ag, As/Sb or Te/In, and in principle the trends are reproduced by the less precise EDXRF results (not shown as figure). However, two of the figurines analysed by ICP-Q-MS do not follow the trends: MA-202522 and −23. They show clear deviations in almost all element combinations with lead isotopes (Fig 6). Therefore, they will not belong to the presumed mixing line, even though they have lead isotope compositions largely matching the line defined by the other bronze objects. Could it thus be that we are dealing not with just one mixing line (hereafter called mixing line 1a, ML1a), but rather two or perhaps more mixing lines (mixing line 1b, ML1b) running parallel or congruently? Finally, a crucial question emerges: do bronze objects from different mixing lines represent raw materials or mixtures from the same or various sources?

### 4.2.2.2. Evidence for various copper pools:

To answer the above questions, we use the copper isotope composition as an additional proxy. Basically, copper isotopes provide information about the ore base, e.g., whether it was a primary or secondary ore, but they can moreover be used for the reconstruction of metallurgical practices such as mixing/recycling, as recently demonstrated [76–80,69,61,70]. In the bronzes most likely belonging to ML1a, $\delta^{65}$Cu values vary only slightly from –0.30 to –0.18‰, but two groups separate: one with four samples with values around –0.28‰, and a second with five samples with values around –0.19‰ (Fig 7c). The two bronze objects seemingly belonging to other mixing lines (MA-202522, −23) display much more negative $\delta^{65}$Cu values down to –1.0‰; the same also applies to two additional *bronzetti* (MA-202519 and-33) ranging down to –1.3‰ (see also Table 4). This finding prompts further refining of the conclusions on mixing reached above.

Firstly, this finding underlines that the arms and feet fragments of ML1a are in fact closely related, in that they were mixed from copper bases that were almost identical in terms of copper isotopes, but distinctly different in terms of the lead isotope and slightly different in chemical composition. Since the copper isotope values, however, seem to separate into two groups, it is very likely that ML1a is further split into two parallel mixing lines, ML1a1 and ML1a2. This means that at least three copper sources or batches of two pools of copper were involved in mixing. This, in turn, explains why we observe overall trends in the data, but not proper alignment for all objects (cf. Fig 13d–e). Assuming, for reasons that will be explained further below, that we know the end members of ML1a – that is, the original materials used for mixing – then we are dealing with two copper bases (Cu1a and Cu1b) of one copper pool (Cu1), whose signatures are:

$$\delta^{65}Cu \approx -0.18 \; to -0.25‰$$

$$^{206}Pb/^{204}Pb = 17.908 - 17.921, ^{207}Pb/^{204}Pb = 15.646 - 15.649, ^{208}Pb/^{204}Pb = 38.010 - 38.021$$

$$Ni \approx 0.03\%, Ag \approx 0.02\%, As \approx 0.25\%, Sb \approx 0.02/0.03\%, Pb \approx 1\%$$

and which are represented best by *bronzetti* MA-202521 and MA-202542, and a further copper base (Cu2a) of a second copper pool (Cu2) with a composition of about

$$\delta^{65}Cu \approx -0.20‰$$

$$^{206}Pb/^{204}Pb = 18.185, ^{207}Pb/^{204}Pb = 15.610 \text{ and } ^{208}Pb/^{204}Pb = 38.320$$

$$Ni \approx 0.03\%, Ag \approx 0.06\%, As \approx 0.08\%, Sb \approx 0.02\%, Pb \approx 4-5\%$$

and which is represented by *bronzetto* MA-202515 (Fig 7a–b; 8).

Secondly, the bronzes MA-202522 and −23 deviating from the other objects were clearly made with different (isotopically lighter) copper. The observed negative δ65Cu cannot have been obtained by mixing of copper from pools Cu1 and Cu2, so that an additional copper base from another copper pool must have been used for their production. Yet it is not possible to narrow down its original composition with the available data in this study. Thirdly, a separate copper must also have been used for the *bronzetti* MA-202519 and −33, due to their strongly negative copper isotope composition (Table 4). This is the most surprising result, as these figurines initially appeared to be representatives of

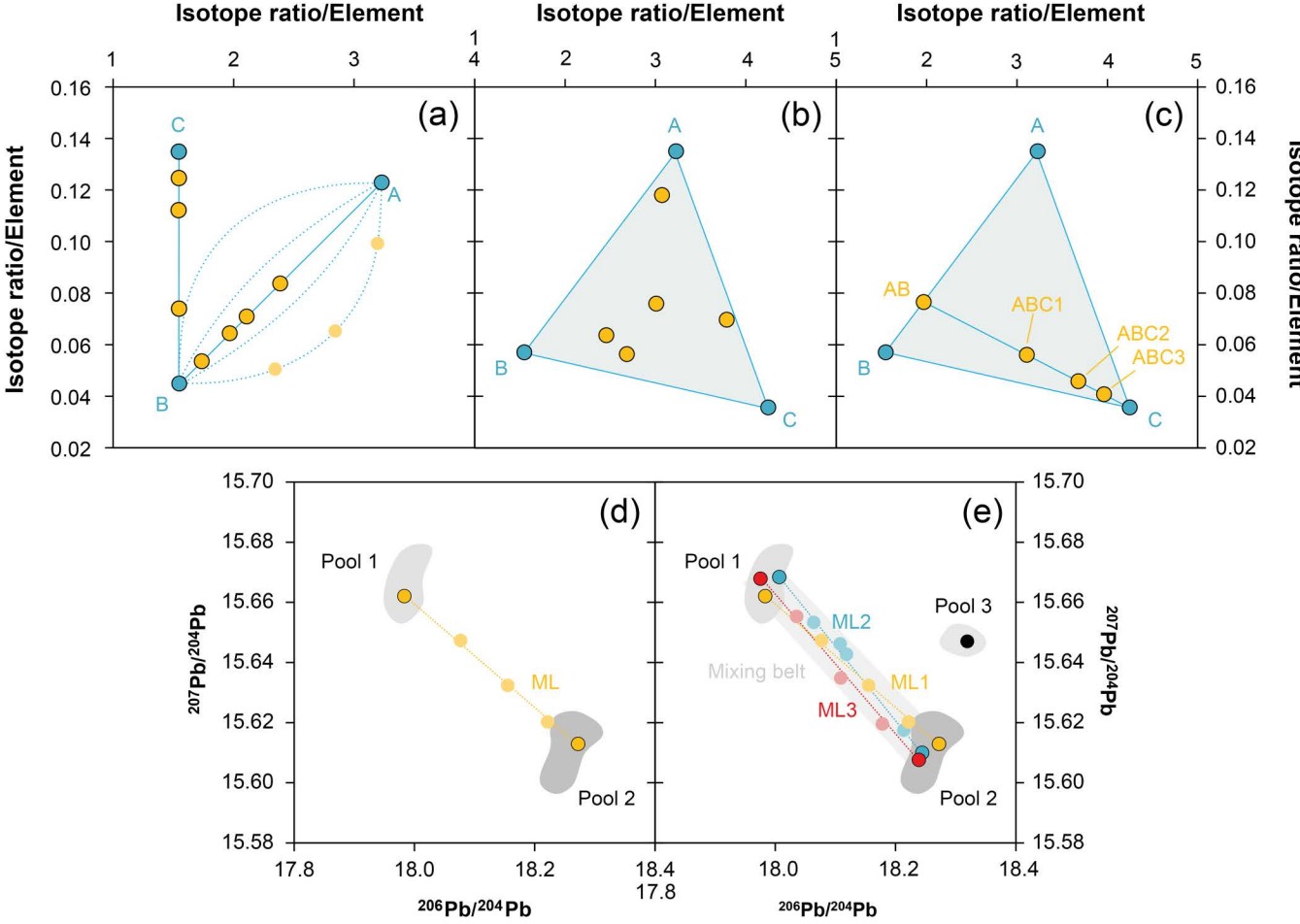

**Fig 13. Theoretical mixing scenarios in metals.** Two-component (a) and three-component mixing (b), illustrating the relationship between the original materials or end members (blue symbols) and the resulting mixtures (orange symbols) positioned along straight or curved mixing lines. In (c), sequential mixing is shown, where after a first mixing of components A and B, a second mixing step of the mixture AB with C resulted in the mixing products ABC. (d) and (e) display emerging mixing lines (ML) between the end members of two pools or stocks. Due to slightly varying signatures within these pools and multiple mixing events over, e.g., an extended period, an overall trend (= mixing belt) with increased data spread is observed, although it is composed of several mixing lines (image: D Berger).

ML1 according to their chemical composition and lead isotope ratios (Fig 8). With their incompatible copper isotopy, this interpretation is no longer valid, and there could indeed be more than one or two mixing lines existing in parallel, while leading to a bunch of mixing lines resembling a mixing belt (Fig 13d–e). Nevertheless, it is still striking that both objects are almost identical in their lead isotope ratios (as already stated above as cluster) and trace-element pattern to Cu2 (MA-202515), which has been proposed as an end member of ML1a. This allows only two possibilities: (1) either copper with natural impurities of lead from the same source with the same lead isotope but different copper isotope composition was used, or (2) copper from different or the same deposits was intentionally alloyed with lead. Since the lead concentration of these bronze objects is consistently high at about 4–5%, and such levels are usually perceived as manmade, there is a strong case for the second possibility. However, for a proper decision, the data of the *bronzetti* must be assessed together with the lead isotope data from concrete ore deposits and the existing evidence of their exploitation

### 4.2.2.3. Provenance of copper pool Cu2:

Despite the above efforts, the intentional addition of lead to copper remains a possible option. The data of the Santa Vittoria *bronzetti* are thus compared here with both copper and lead mineralisation in the Mediterranean (Fig 14), while non-matching regions are dismissed (e.g., the Alpine region, southern France, Great Britain, or important lead mines like Cartagena, Spain, and Lavrion, Greece). It is noticeable in the plots that no ores compatible with the Cu2 signature occur in Sardinia; hence, exogenous lead or copper must have been used. In fact, ores with matching lead isotope ratios are rather to be found in the Iberian Peninsula; those with the best match in all isotope ratios are located in the Alcudia valley mineral field of the Sierra Morena in the Central Iberian zone (see Table 6). Together with the adjacent Linares–La Carolina and Los Pedroches areas, the Alcudia valley was an important supplier of metals in the nineteenth to twentieth centuries CE, especially the large lead-zinc deposits, with galena making up 95% of the mineralisations [81,87–89]. However, there is indication that mineral resources, even copper, were mined in this region as early as the Bronze Age [88,90–93]. This makes the ores in the Alcudia valley currently the most likely source supplying lead or copper for Cu2, which was used as one of at least three copper pools for the Santa Vittoria *bronzetti*.

This conclusion is substantiated by recent findings from Iberia. Montero Ruiz [88] and Montero Ruiz et al. [94] published analytical results comprising bronze objects from Iberian Late Bronze Age hoards with similar lead isotope ratios to the Santa Vittoria bronze objects. These objects are consistently low in lead (<1%) while compatible with the Alcudia valley ores, and so are argued to have been made with copper from the rare copper deposits in the mineral zone, two of which are the mining districts of El Garbanzal and Brazatortas [88]. Neither indications of ancient exploitation nor chemical and isotopic data is yet available from the two locations. However, if they were indeed used, the ores from these two sites could have supplied copper for artefacts from the western Iberian Peninsula, given the matching isotope signatures with the Alcudia valley mineralisation.

Nevertheless, copper ores from the Linares mining district are possible sources for Cu2 copper, based on their very similar lead isotope ratios (Fig 14a–b). What could make them more likely than ores from the Alcudia valley is the fact that the Linares signatures are more frequently observed in metal artefacts from the eastern Iberian Peninsula, while the Alcudia valley ores were apparently more often used for objects in the western regions [88]. Regardless of whether Alcudia valley or Linares ores were used, the important implication for the figurines studied is that it is still possible that copper from Central Iberia was used for their production, and that this copper contained natural impurities of lead of up to 5%. Yet alloying with lead cannot be excluded categorically; and if so, this lead must have come from the same mining regions.

### 4.2.2.4. Origin of copper pool Cu1:

This leads to the provenance of copper pool Cu1, which can be perceived as the second end member of ML1a. This contains about 1.5% lead, but unlike Cu2, this concentration can well be explained by the natural association of lead and

copper minerals/ores. Smelting experiments with oxide ores have shown that even if a significant amount of lead is lost, large quantities of it pass from the ores into the copper [95]. On these grounds, Gale [1] argued that there is no reason to assume intentional lead addition to Sardinian copper at lead contents of 2%. Yet his reasoning is only valid if source

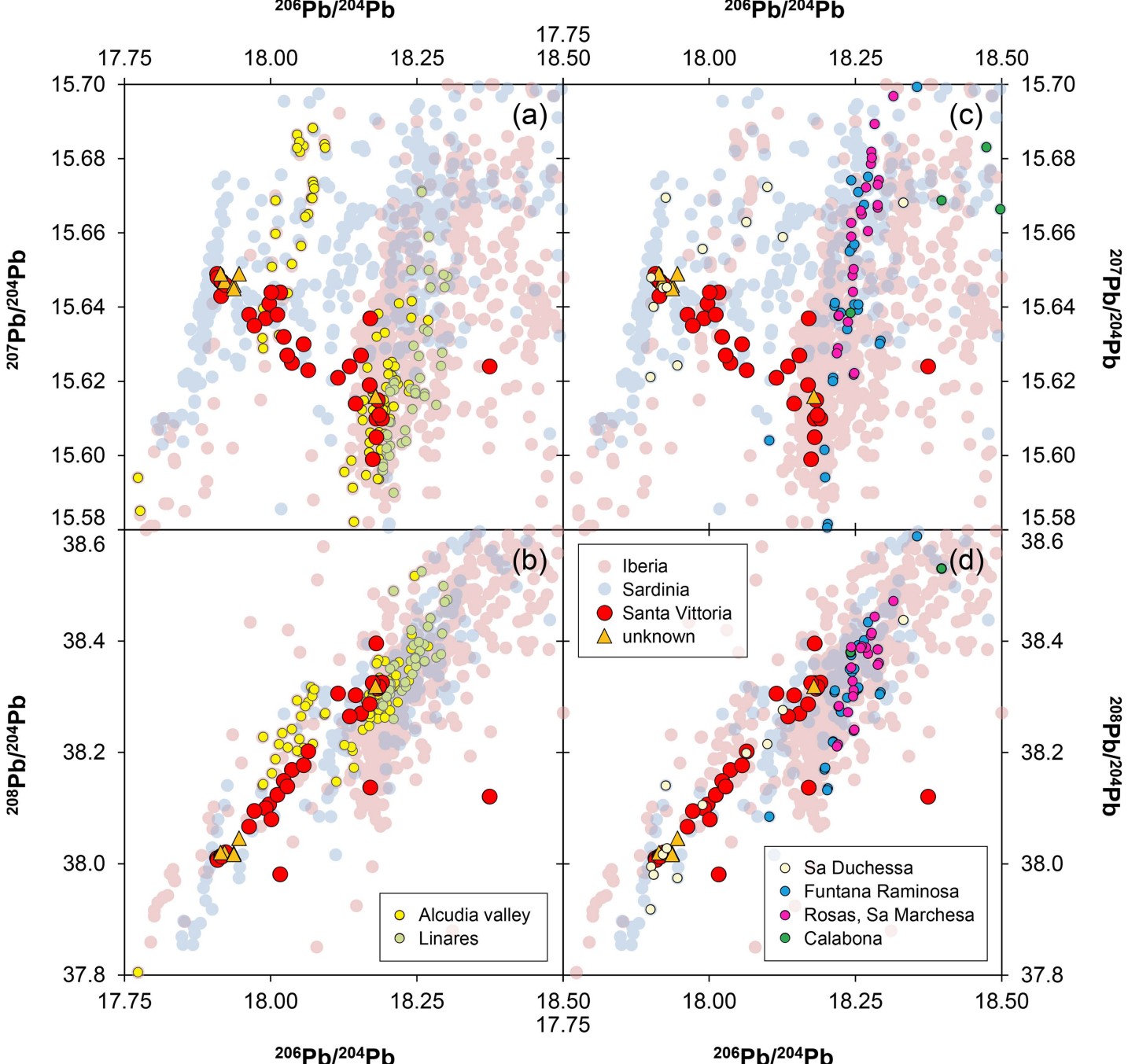

**Fig 14. Comparison of lead isotope ratios of the figurines from Santa Vittoria and the unknown Sardinian site with copper and lead ores from Sardinia and Iberia (diagrams: D Berger; data from ingots and ores from references given in S4 Appendix).**

mixing or recycling can be ruled out. For example, if a copper containing a high amount of intentionally alloyed lead (e.g., 5%) was mixed with a naturally low-lead copper (e.g., 1%), the mixtures could contain 2% or even less lead. In such a case we would classify the low-lead concentration as a natural impurity of the copper [1,56], while in reality it would be a mixture of natural and intentionally added lead. This once again relativises the elemental concentration as a rigid decision

Table 6. List of lead isotope ore reference data from the literature showing the best matches with the objects of this study identified as possible end members of mixing lines.

| Sample no. | Region | Deposit/mine | Elements | $^{208}$Pb/$^{206}$Pb | $^{207}$Pb/$^{206}$Pb | $^{206}$Pb/$^{204}$Pb | $^{208}$Pb/$^{204}$Pb | $^{207}$Pb/$^{204}$Pb | Reference |
|---|---|---|---|---|---|---|---|---|---|
| MA-202515 | Santa Vittoria | | | 2.1081 | 0.85866 | 18.180 | 38.325 | 15.610 | This study |
| ARQ-1A | Linares–La Carolina | Grupo Minero Arquillos | Pb-Zn | 2.1047 | 0.8574 | 18.192 | 38.289 | 15.598 | [81] |
| 835–088 | Valle del Alcudia | El Horcajo | Pb-Zn | 2.1048 | 0.8578 | 18.185 | 38.276 | 15.599 | [81] |
| 835–085 | Valle del Alcudia | Mina Encarnación | Pb-Zn | 2.1052 | 0.8578 | 18.188 | 38.289 | 15.602 | [81] |
| 836–012 | Valle del Alcudia | Mina Diógenes | Pb-Zn | 2.1030 | 0.8580 | 18.194 | 38.262 | 15.606 | [82] |
| 835–093 | Valle del Alcudia | Los Ángeles | Pb-Zn | 2.1061 | 0.8582 | 18.185 | 38.299 | 15.606 | [81] |
| 836–034 | Valle del Alcudia | El Encinarejo | Pb-Zn-Ag | 2.1055 | 0.8585 | 18.184 | 38.286 | 15.611 | [81] |
| 835–092 | Valle del Alcudia | La Veredilla | Pb-Zn | 2.1070 | 0.8584 | 18.187 | 38.320 | 15.612 | [81] |
| 835–039 | Valle del Alcudia | La Emperatriz | Pb-Zn | 2.1068 | 0.8590 | 18.175 | 38.291 | 15.612 | [81] |
| 835–030 | Valle del Alcudia | San Rafael | Pb-Zn | 2.1059 | 0.8582 | 18.192 | 38.311 | 15.612 | [81] |
| GAL-38 | Valle del Alcudia | El Peñoncillo | Pb-Zn | 2.1079 | 0.8585 | 18.186 | 38.334 | 15.613 | [81] |
| 836–079 | Valle del Alcudia | Mina Villalba | Pb-Zn | 2.1054 | 0.8586 | 18.185 | 38.287 | 15.614 | [81] |
| 835–143 | Valle del Alcudia | La Salvadora | Pb-Cu-Ag | 2.1055 | 0.8582 | 18.194 | 38.307 | 15.614 | [81] |
| 835–007 | Valle del Alcudia | La Jarosa | Pb-Zn | 2.1058 | 0.8583 | 18.193 | 38.311 | 15.615 | [81] |
| 835–011 | Valle del Alcudia | Joffre | Pb | 2.1080 | 0.8590 | 18.174 | 38.306 | 15.616 | [82] |
| MA-202521 | Santa Vittoria | | | 2.1226 | 0.87389 | 17.908 | 38.010 | 15.649 | This study |
| 22 | Iglesiente-Sulcis | San Benedetto | Zn-Fe | 2.1224 | 0.8737 | 17.905 | 38.002 | 15.643 | [83] |
| SD 1006D1 | Iglesiente-Sulcis | Sa Duchessa | Cu | 2.12137 | 0.87301 | 17.921 | 38.017 | 15.645 | [46] |
| SBND4 | Iglesiente-Sulcis | San Benedetto | Zn-Fe | 2.12251 | 0.87364 | 17.911 | 38.016 | 15.648 | [84] |
| IG49 | Iglesiente-Sulcis | Campo Pisano (G. Luas) | Zn | 2.12164 | 0.87361 | 17.913 | 38.005 | 15.649 | [85] |
| SED17 | Iglesiente-Sulcis | Seddas Moddizzis | Pb-Zn | 2.12089 | 0.87342 | 17.917 | 38.000 | 15.649 | [85] |
| MA-202501 | Su Monte | | | 2.1224 | 0.87391 | 17.904 | 38.000 | 15.647 | This study |
| COR2 | Iglesiente-Sulcis | Coremo | | 2.12304 | 0.87436 | 17.898 | 37.998 | 15.649 | [84] |
| CARM | Iglesiente-Sulcis | Monteponi | Pb | 2.12267 | 0.87398 | 17.902 | 38.000 | 15.646 | [85] |
| | Iglesiente-Sulcis | San Benedetto | Zn-Fe | 2.12242 | 0.87367 | 17.905 | 38.002 | 15.643 | [83] |
| ARI05 | Iglesiente-Sulcis | Tiny Arenas | Pb-Zn-Ba | 2.12277 | 0.87388 | 17.903 | 38.004 | 15.645 | [85] |
| SUZU6 | Iglesiente-Sulcis | Su Zurfuru | Pb | 2.12242 | 0.87418 | 17.906 | 38.004 | 15.653 | [85] |
| IG49 | Iglesiente-Sulcis | Campo Pisano (G. Luas) | Zn | 2.12164 | 0.87361 | 17.913 | 38.005 | 15.649 | [85] |
| MA-202504 | Su Monte | | | 2.1028 | 0.85651 | 18.289 | 38.458 | 15.664 | This study |
| | Iglesiente-Sulcis | Mont'Ega | Pb-Zn | 2.10240 | 0.85621 | 18.291 | 38.455 | 15.661 | [83] |
| | Iglesiente-Sulcis | Mont'Ega | Pb-Zn | 2.10203 | 0.85627 | 18.298 | 38.463 | 15.668 | [86] |
| SARD 100-B | Iglesiente-Sulcis | Bruncu Lionaxi | | 2.0989 | 0.8559 | 18.298 | 38.406 | 15.661 | [6] |
| PS 1013E | Iglesiente-Sulcis | Pranu e'Sanguini | | 2.10007 | 0.85619 | 18.296 | 38.423 | 15.665 | [84] |
| SM 1001G | Iglesiente-Sulcis | Sa Marchesa | Pb-Zn-Cu | 2.09894 | 0.85667 | 18.288 | 38.385 | 15.667 | [84] |
| | Iglesiente-Sulcis | Sa Marchesa | Pb-Zn-Cu | 2.09890 | 0.85670 | 18.288 | 38.385 | 15.668 | [46] |
| MA-202505 | Su Monte | | | 2.1138 | 0.86636 | 18.066 | 38.189 | 15.652 | This study |
| | Iglesiente-Sulcis | Malacalzetta | | 2.11404 | 0.86657 | 18.055 | 38.169 | 15.646 | [83] |
| | Iglesiente-Sulcis | Terras Nieddas | | 2.11192 | 0.86557 | 18.076 | 38.175 | 15.646 | [83] |
| SARD 129b | Iglesiente-Sulcis | Domusnovas | | 2.1147 | 0.8671 | 18.064 | 38.199 | 15.663 | [6] |

criterion for intentional lead addition. Thus, hypothetically, one must assess in each individual case whether the lead in copper or bronze has been added intentionally or not – which, of course, is often not easy or impossible in practice.

In the case of Cu1, there is indeed no reason to assume an intentional addition of lead. Firstly, the lead concentration is no more than 1.5%; secondly, the lead isotope signatures of this and other samples in our suite form a tight cluster plotting onto the isotopic field of Sardinian mineralisation (Fig 14c–d). This data belongs to ores or minerals from the so-called 'metalliferous ring' in the Iglesiente-Sulcis region in southwest Sardinia. This area is rich in lead-zinc mineralisation such as those at Montevecchio, Monteponi or San Giovanni (Fig 1), but it also hosts polymetallic ore bodies in which primary and/or secondary copper minerals are associated with lead, zinc and silver minerals [18,96,97]. Smelting such ore assemblages inevitably produces impure copper, containing appreciable amounts of lead and other elements [95]. An origin of Cu1 from ores in the Iglesiente region is also supported by its chemical composition, since all impurities (Ni, As, Ag, Sb, Pb) are widespread in the ores of the area. Of the large number of ore samples analysed so far, those summarised in Table 6 are the most compatible with the Santa Vittoria bronzes. They belong to the mines of San Benedetto (Zn-Fe), Seddas Moddizzis (Zn) and Campo Pisano (Zn-Pb), all of which were of no interest to ancient copper miners, as copper minerals are minor components there. In contrast, one sample from the Sa Duchessa mine is interesting (Fig 1), as secondary copper minerals (mainly malachite and chrysocolla) occur massively in addition to primary copper, lead, and zinc ores [18,46,77,98–103]. The lead isotope ratios of other samples from the mine (including samples from Domusnovas, the town in whose territory Sa Duchessa is located) are close to the mentioned sample. However, with only a few samples, it is currently not possible to reliably conclude an origin from the mine based only on lead isotopes. Furthermore, as Valera et al. [18] stated, the isotopic composition of Sa Duchessa is hardly distinguishable from that of other Iglesiente mineralisation. However, other relevant and partially closer copper deposits such as Funtana Raminosa (closer), Calabona, Baccu Locci, Rosas-Sa Marchesa (Fig 1) differ significantly from the studied bronzes (Fig 14c–d), so that Sa Duchessa is currently the best candidate as source for copper from pool Cu1. Previous studies have already argued that this mine (more broadly subsumed as 'Domusnovas') supplied both copper and lead for metalwork [46,47]; the match between artefacts and ore/mineral samples, however, has never been as close as in the present study.

**4.2.2.5. Disentangling overlap with Levantine copper ores by osmium systematics.**
An overlap exists between the lead isotope composition of Sa Duchessa (and Sardinian mineralisation in general) and copper ores in the Arabah valley (for example the mining districts of Faynan, Jordan, and Timna, Israel: Fig 15a–b). Moreover, ancient exploitation in the Arabah is variously attested, as well as connections between Sardinia and the Levant in the late Nuragic and Phoenician periods [47,78,100–108]. Due to its chemical characteristics with very low silver, antimony, and bismuth, Faynan is not a likely source for the *bronzetti* (Fig 15c). Ores from Timna, however, display similar isotopic and chemical features to those observed in Sardinian mineralisation (Fig 15). In particular, copper isotope values are as negative as those in some figurines [78,109–112]. Therefore, the mining district of Timna cannot categorically be excluded as source region for the *bronzettis*' copper. The analytical proxies employed hitherto render a proper differentiation difficult. For this reason, osmium isotope ratios were included as a new proxy by the present study, not least to explore whether the copper from the two copper pools, Cu1 and Cu2, and thus their potential source regions in Sardinia, the Arabah and the Iberian Peninsula, can be distinguished independently of the lead isotope ratios.

The figurines from Santa Vittoria have extremely radiogenic $^{187}Os/^{188}Os$ ratios, with the highest one observed for MA-202521 (Table 5) from copper pool Cu1. At the same time, this sample has a very high osmium concentration of 5000 pg g$^{-1}$. Osmium isotope ratios and concentrations are not yet available from Sardinian ores, but from copper, iron, and manganese ores from Jordan and Israel, as well as from copper and iron artefacts. The Jordan ores consistently show $^{187}Os/^{188}Os$ ratios below 9 [113,114], which also applies to copper objects from the Negev desert (Fig 16 and unpublished data CEZA and N Yahalom-Mack). As the osmium contents of the copper ores are very low, these alone cannot have been the source for Cu1 and mixtures thereof. This interpretation does not change if fluxing with iron or manganese ores from Faynan or the wider region (e.g., the northern Negev desert) during smelting is taken into account, materials that

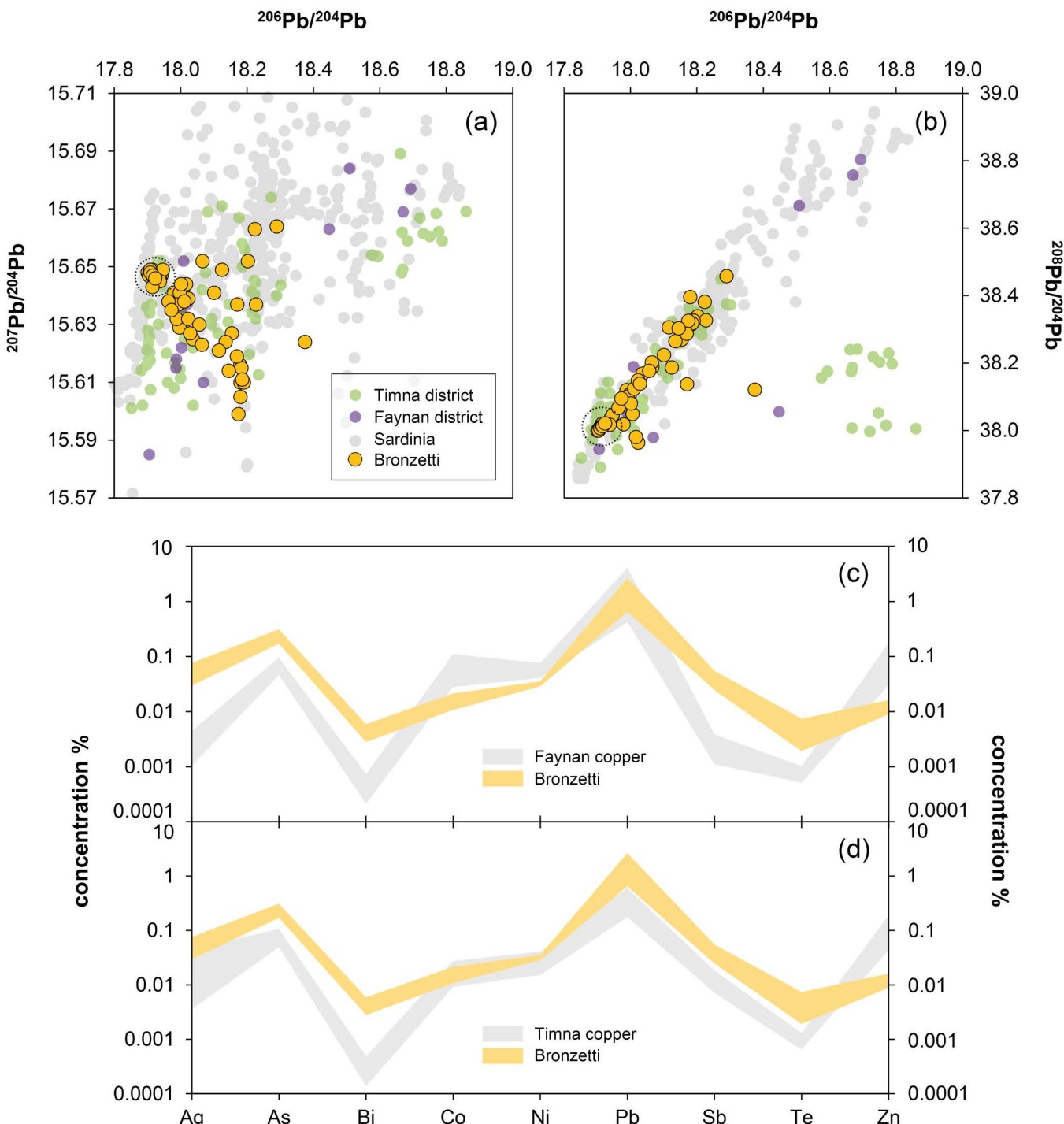

**Fig 15. Lead isotope ratios of Sardinian ores and *bronzetti* as well as copper ores from Timna and Faynan in the Arabah.** In (a–b) the dotted circle indicates the isotopic range with the largest overlap between Sardinia and Timna. Also shown is the comparison between the elemental composition of copper of the Timna and Faynan mining districts (minimum and maximum values) and the Sardinian figurines (c–d) (diagrams: D Berger; data from ingots and ores from references given in S4 Appendix).

are considered to be the hosts of the osmium in the Levant. This practice has been demonstrated for Faynan, Timna, and other sites of the region [115–117] and might be an explanation for elevated osmium concentrations in the smelted copper, but not for the extreme $^{187}Os/^{188}Os$ ratios. Iron ores from Timna and the close Nahal Amram, as well as Timna and Nahal Amram copper, for their part, show high osmium concentrations, but very low $^{187}Os/^{188}Os$ ratios (Fig 16). Hence, Timna copper seems as incompatible with the Cu1 copper of the *bronzetti* as the material from Faynan. This leads the focus back to the copper deposits of Sardinia, and the Iglesiente-Sulcis district in particular, as they host extensive molybdenum mineralisation, with molybdenite ($MoS_2$) as the chief mineral [97,118–120].

Rhenium and osmium isotopes have been used for decades to date the formation of molybdenite, a common mineral in ore deposits and the world's main source of molybdenum and rhenium. The isotope $^{187}Re$ decays via beta decay to $^{187}Os$. Due to the extremely high Re-Os ratios in molybdenite, or even the complete absence of common osmium, the osmium isotopic signature in molybdenite is always characterised by an extremely high $^{187}Os/^{188}Os$ ratio, making it highly radiogenic. Notably, molybdenite is frequently observed in the polymetallic ore deposits of Sardinia [97], and has even been described from Sa Duchessa (https://www.mindat.org/loc-2128.html). Smelting the oxidised ores from this mine, which might have contained remnants of molybdenite or weathering products thereof, would likely transfer the highly radiogenic signature to the copper. We therefore consider molybdenite to be the most likely source for the extreme osmium isotope signature and high osmium concentrations in a portion of the Santa Vittoria bronze objects, which in turn would prove a Sardinian origin of the copper.

Yet the results also allow additional conclusions. The oxidation products of osmium ($OsO_4$) are highly volatile, so any smelting operation under an oxidising atmosphere would result in a high osmium loss. Thus, the roasting of sulphide ores such as chalcopyrite can virtually be excluded, as this would have significantly lowered the osmium content in the copper, which is obviously not the case given the high osmium concentrations in the figurines (Table 5). The data therefore

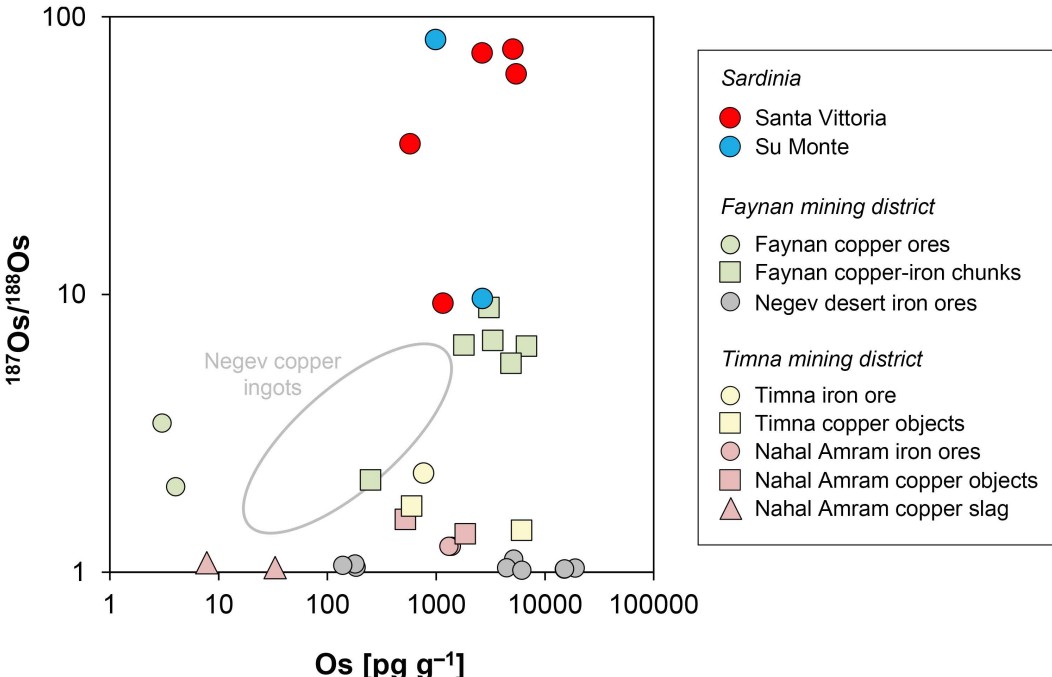

**Fig 16. Osmium concentrations and osmium isotope ratios of the analysed *bronzetti* and copper ingots from this study and copper and iron ores and objects from the Arabah valley and the Negev desert.** The grey ellipse represent copper ingots of unknown provenance in the Negev (diagram D Berger, I Stepanov, M Brauns; data from CEZA and [113,114]).

suggests that a secondary oxide ore, such as malachite or chrysocolla, was used for the production of Cu1, which again is a strong indication for the Sa Duchessa ores. The copper isotope values of the Santa Vittoria bronzes, which are principally a measure of the ore type used [77,76,121], do not disqualify this conclusion. Even though, copper isotope data is not yet available for Sardinia, comparable negative values to those in the figurines were reported in oxidic copper ores from elsewhere [78,109,111]. Equally importantly, previous trials to apply the Re-Os chronometer to galena failed due to the absence of Re and Os in this lead mineral; hence galena and smelted lead can most likely be excluded as source for the osmium and its high radiogenic ratios. Together with other indications, this proves that Sardinian copper, not copper from elsewhere with intentional lead addition, was used as copper from pool Cu1. Nevertheless, these are preliminary considerations, as roasting/melting experiments and analyses of Sardinian copper ores (including copper isotopes) must follow, which are underway.

Regardless, it is interesting that the osmium isotope ratios of the other end member of ML1 (Cu2 and comparable) compatible with Iberian ore source are significantly lower and less radiogenic at 9–35 (Fig 9). This is an important observation. It is further indication of a principal difference in the ore bases for Cu1 and Cu2 and different supply regions. Moreover, it underscores that the analysed samples MA-202515 ($^{187}Os/^{188}Os = 35$) and MA-202533 ($^{187}Os/^{188}Os = 9$) can by no means be part of the same mixing lines. Principally, osmium could support the above interpretation regarding source mixing (tendencies are observable with lead isotopes (Fig 9b); but the current sample suite is not large enough for statistically robust statements. The allocation of objects to different mixing lines (e.g., MA-202533 vs. MA-202515) and the possible loss of osmium by copper melting and refining (without changing the isotope ratios!) affect meaningful conclusions.

**4.2.3. Su Monte.** The trend in the lead isotope composition from low ($^{206}Pb/^{204}Pb = 18.000$, $^{207}Pb/^{204}Pb = 15.640$) to more radiogenic ratios ($^{206}Pb/^{204}Pb = 18.290$, $^{207}Pb/^{204}Pb = 15.665$) seems to suggest the existence of a mixing line (hereafter ML2) also for the *bronzetti* from Su Monte (Fig 7a–b). It even appears possible that there is an intercept with ML1. However, the trace-element concentrations and their combination with the lead and copper isotopes reveal that we are, rather, dealing with at least two groups of bronze objects (Fig 8). The observed overall trend is thus only apparent.

Within one of these groups, and based on the data, several objects can clearly be related with each other. The bronzes MA-202501, MA-202502 and MA-202503 show linear alignment not only in the lead isotope ratios (mixing line ML3, Fig 8a), but also in most combinations with trace elements and copper isotopes (Fig 8). Although only three samples are available, the correlations clearly indicate a mixing of sources. Because the copper isotope values change with changing lead isotope ratios, mixing of two different copper bases is very likely. With about 6%, one of the samples (MA-202501) could contain intentionally added lead. Its lead isotope composition is very similar to that of Cu1 of ML1 (Santa Vittoria), and hence, again, consistent with Sardinian ores; the best matches are with samples from the lead mines of Monteponi, San Giovanni and San Benedetto (Table 6; Fig 17). However, in principle, the same source as for Cu1 is also possible, not least because Sa Duchessa has extensive lead mineralisation in addition to copper ores. Comparison with Nuragic lead objects illustrates that the full range of Iglesiente lead ores (and beyond) were used at that time, and different mines were supposedly in operation. Of these lead artefacts, a few have similar isotopic compositions to MA-202501 (Fig 17c–d). Overall, the results illustrate that an (intentionally leaded?) copper base was mixed with a second copper source, whose origin cannot be reconstructed due to the unknown end member of the corresponding ML3. According to the copper isotope composition, a similar copper source as for the bronze objects from Santa Vittoria was likely used for MA-202501 ($\delta^{65}Cu = -0.31 \pm 0.03‰$), while for MA-202503 a copper rich in secondary sulphides or oxides thereof could have been used, indicated by the strongly negative $\delta^{65}Cu$ value ($-0.89 \pm 0.01‰$).

The above statements cannot be transferred to the other five bronzes from the sanctuary. Neither the trace elements nor their combination with lead or copper isotopes allows a clear conclusion as to whether the objects were related (Fig 5–6, 8). Some elements could correlate, but as this cannot be applied to all the objects, a direct relationship is unlikely or non-existent. However, we have shown in the previous section that several mixing lines can exist in parallel, depending on the number of batches mixed from two copper pools (Fig 13d–e) – but also that conclusions about that can be

difficult when the number of samples is limited. The situation is even more complicated because one bronze (MA-202504) contains 16% of intentionally added lead. This sample situates towards the more radiogenic part of the trend in the lead isotope ratios, while the lead concentrations progressively decrease with decreasing ratios in the other samples. This observation still does not provide a clue as to mixing, but it opens up the possibility that the observed trend is dominated by the isotopic signature of the alloyed lead. From the isotopic data, it becomes clear that this lead component, and the lead used for mounting one *bronzetto* to a base (MA-202505) are different. Even though we did not analyse the corresponding bronze, the result suggests that the production of the copper-lead alloy and the casting of the figurines were likely not coupled to the mounting process involving lead in the sanctuary.

Should the lead isotope composition of the five bronzes be the consequence of source mixing, the reconstruction of the origin of the raw materials will be difficult, as is often the case when sources were blended. Firstly, we cannot be sure about this practice for the five bronze objects; secondly, no clustering appears in the diagrams, due to the low number of samples (in contrast to Santa Vittoria). The original composition of the potential end member is thus unknown, affecting conclusions on provenance. Yet provided there is no direct relationship between the individual bronzes, Sardinian sources are as possible as Iberian and French deposits for most items (Fig 17). Overall, the match with Sardinian lead and copper ores from the Iglesiente-Sulcis district is better; but sources in France (not shown) and Iberia cannot be properly excluded because of the general large data overlap in the isotopic region of the bronzes. On the other hand, ores from Cyprus, Greece, the southern and western Alps, Tuscany, and Great Britain, as well as the Sinai and Levant, can largely be ruled out due to the mismatch of data (not shown). The most secure statement can be made for MA-202504 with its 16% lead, as the isotopic signature is most likely dominated by the deliberate addition of lead. Accordingly, the added lead shows the most convincing match with lead ores from the Mont'Ega mine in the Iglesiente-Sulcis region of Sardinia. But other sites, such as those from the Rosas-Sa Marchesa mineralisation zone with frequent occurrences of copper ores, would also be possible (Table 6; Fig 17) in situations when the lead was introduced, unwanted, along with the copper. Thus, the use of local copper and lead ores is supported at least for some of the Su Monte *bronzetti*.

For further discrimination, osmium isotope ratios should be used. Unfortunately, unlike Santa Vittoria, only one *bronzetto* (MA-202506) and one copper ingot (MA-202494) from Su Monte could be analysed (Table 5; Fig 7e). Overall conclusions can thus not be drawn. Both datasets are nevertheless revealing. The copper of the ingot matches the Cu1 copper pool with *bronzetti* MA-202521 and MA-202542, and the Sa Duchessa ores, in the lead isotope ratios. At the same time it has a comparably high $^{187}Os/^{188}Os$ ratio of 83. This result underscores that Sardinian copper from the Iglesiente-Sulcis district stands out for extraordinary osmium isotope systematics, and that local copper (only 2% Pb in the ingot; Table 2) was used in ingot form and most likely distributed across the island. The $^{187}Os/^{188}Os$ ratio of 10 of the Su Monte *bronzetto* (MA-202506) in turn is as high as that of *bronzetto* MA-202533 from Santa Vittoria, for whose copper an Iberian origin was considered. Possibly, the elevated osmium isotope ratio and composition of the piece from Su Monte can still be tracked to ores in the Iglesiente-Sulcis district as the lead isotope ratios overlap with those from Sardinia. However, since copper mixing likely happened, the $^{187}Os/^{188}Os$ ratio originally could have been much higher depending on the characteristics of the raw materials. To interpret this result, systematic collection of ore and artefact data from various regions is required.

**4.2.4. Abini and the unknown site.** The same considerations as above apply to the *bronzetti* from the unknown site. The lead isotope composition of five out of the six samples (MA-202497, MA-202510, −11, −13, −14) indicates a production from Sardinian ores, which were certainly mined in the Iglesiente-Sulcis region (Fig 7 and 14). Their striking similarity with the bronzes from Santa Vittoria, especially with Cu1 (MA-202521 and −42) and nearby items, is remarkable, as the lead isotope ratios of the objects form a common cluster around the presumed end member of ML1 (= Cu1). In addition, the chemical composition agrees very well with that of the Santa Vittoria bronzes, as do the copper isotopes (Fig 8). From these results, one can conclude the following. Firstly, the ancient metallurgists often resorted to ores and mineral species from the same deposit for the production of bronze objects, as shown by items with unknown provenance and from Santa Vittoria. Secondly, one can speculate as to whether the metalwork from both places was the result of a

joint *chaîne opératoire*. This assumption is supported by the fact that the lead isotope composition of the sixth sample (MA-202512) from the unknown site is close to Cu2 at the opposite end of ML1 of Santa Vittoria (Fig 8 and 14). Since their chemical compositions also match (only XRF data is available due to the corroded state), and since the reverse lead

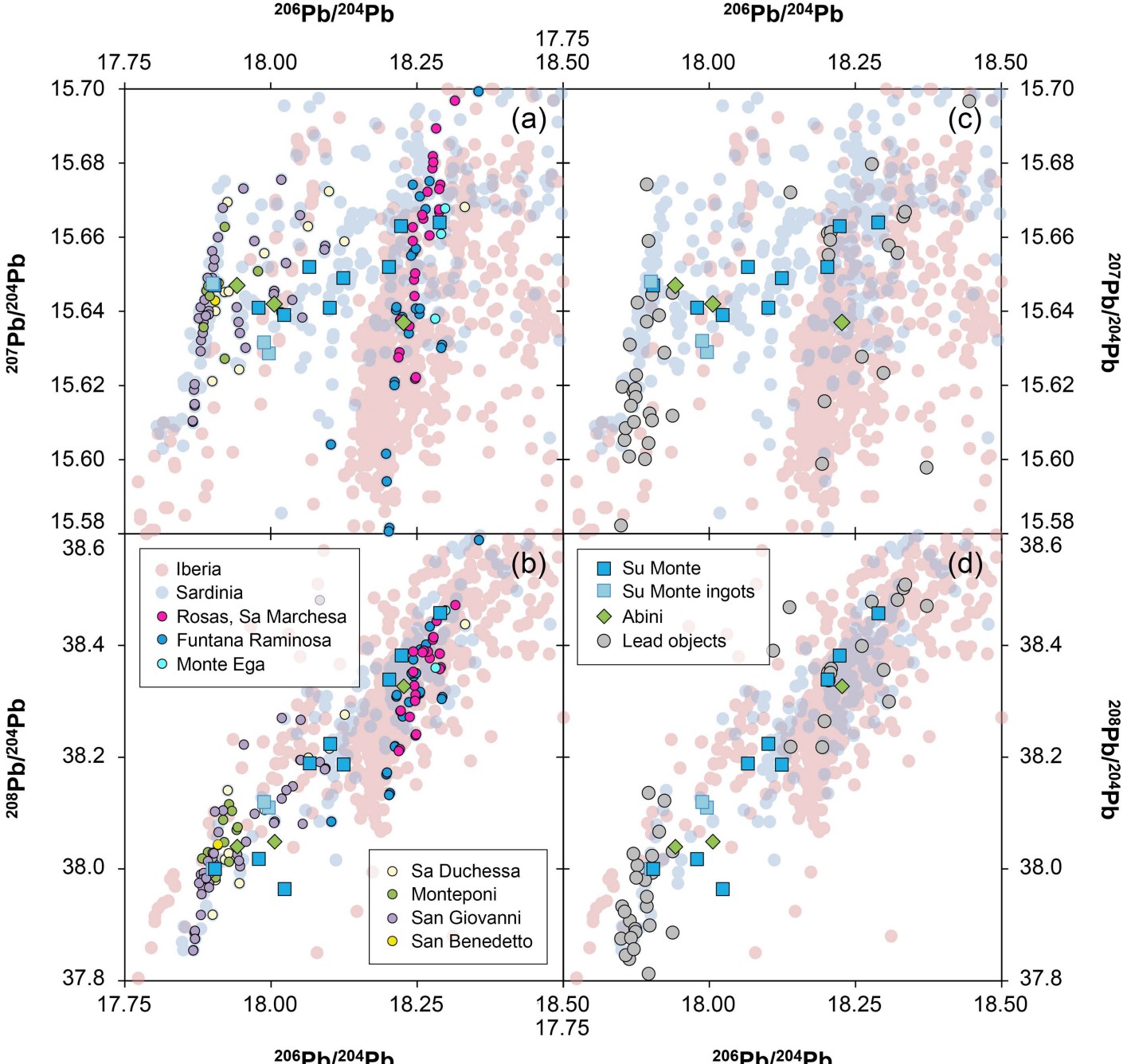

**Fig 17. Lead isotope composition of the figurines and ingots from Su Monte and Abini compared with copper and lead ores from Sardinia and Iberia and Nuragic lead objects (diagrams: D Berger; data from ingots and ores from references given in S4 Appendix).**

content (1/Pb) as well as the lead isotope ratios also correlate perfectly (Fig 6), there is high likelihood that these bronzes belong to the same or a parallel mixing line as the Santa Vittoria *bronzetti*. If this is true, it would imply that objects from different locations were made in a single workshop, which distributed metalwork throughout Sardinia.

In contrast, the copper of the Abini bronzes is different. Although one object (MA-202500) has a similar lead isotope composition to the bronzes from the unknown site and from Santa Vittoria, its chemical composition differs in that it has much less arsenic and antimony, as well as the highest levels of indium and zinc of the entire sample series (Table 4). These observations do not fundamentally rule out the same copper source (e.g., Sa Duchessa), but they indicate that a different ore base or copper batch was used for its production. However, as far as the copper isotopes are concerned, the object is comparable to other bronzes; hence a similar mineral species was likely smelted. The other two objects (MA-202499, MA-202509) stand out due to deviating lead isotope compositions, which appear to establish a negative trend together with MA-202500. Since the trend is not readily reproduced by the trace-element concentrations and their combination with the lead isotope ratios, we can neither exclude nor prove that the Abini bronzes are mixtures of different coppers. However, at least, the *bronzetto* fragment MA-202509 situates very close to an isotopic data cluster of copper ores from Funtana Raminosa and Rosas-Sa Marchesa (Fig 17a–b), which could suggest an origin from these mining districts. Whether this is really the case will only be possible to decide when a larger assemblage from the site is examined.

## 4.3. Mixing copper or mixing bronze?

Previous sections dealt exclusively with the copper of the *bronzetti*, its origin, and the aspect of copper mixing. Nothing has yet been said about the tin and its origin. Nor has mention been made of whether the tin component of the bronzes was already present before the copper was mixed, or whether it was added to the copper mixtures afterwards. This is an important question, though. It has great significance for the interpretation regarding metal trade and processing. On the one hand, the reconstruction of the tin origin could be drastically complicated if the tin isotope signature was the result of a mixing process; on the other, if bronze but not copper was mixed, this could indicate that we are dealing not just with the mixing of materials, but with the reuse and recycling of bronzes. This in turn would shed light on the organisation of metal production in Late Bronze and Early Iron Age Sardinia.

Generally, the reconstruction of bronze mixing follows the same reasoning as for the mixing of copper sources. However, we must take the tin isotope composition and the tin content of the objects into account and compare these parameters with the other proxies, namely the lead and copper isotopy and the chemical composition. In Figs 8g–i (and Fig 7d), the data is summarised and provides a revealing and distinct picture. If bronzes were mixed, the tin isotope composition should in theory correlate with the tin content (1/Sn) of the samples belonging to the mixing lines established above, but also with the trace elements and the other isotope systems. This is actually not the case. For example, the bronze objects MA-202501, MA-202502 and MA-202503 from Su Monte show a very good correlation in most parameters (Fig 8), but no relationship between these parameters is observed, either with the tin concentration or the tin isotope composition. The same is true for the bronzes from Santa Vittoria (Fig 8). With these results at hand, we can exclude the possibility that bronzes or bronze batches were mixed to make the *bronzetti*, and it is equally unlikely that bronze was mixed with copper. The latter would result in a relationship in which the tin isotopy remains constant with changing copper isotope values, as well as tin and trace element concentrations (Fig 13a).

The option of mixing bronze ingots was considered to be a possibility by Begemann et al. [6] for other Sardinian bronzework because they saw the rather constant tin content of the bronzes as indication of that practice. Although we cannot test their hypothesis for the bronze artefacts analysed at that time due to the lack of tin and copper isotope data, our findings make it meanwhile less likely that Late Bronze and Early Iron Age bronzes in Sardinia were made by mixing bronze ingots. Moreover, recycling of bronze objects is also unlikely. Nevertheless, it cannot basically be ruled out, because we have analysed just a small fraction of Sardinian metalwork. More data will follow in due course.

Begemann et al. [6] also posed the possibility that the lead component in Sardinian bronzes could have been introduced along with the tin. This is not completely unreasonable, since tin metal with appreciable lead contents or proper tin-lead alloys was widespread in Late Bronze Age Europe [e.g., [122–127]]. However, there is no evidence that the Nuragic people made use of tin-lead alloys. On the one hand, the Late Bronze Age and Early Iron Age tin ingots known so far from the Mediterranean and beyond as the primary base for tin contain virtually no lead, apart from few exceptions from the Uluburun shipwreck [e.g., [60,62,128–131]]. On the other hand, the handful of contemporary tin artefacts from Sardinia are also lead-free, or contain just a small amount of lead (ca. 1%). Alloying this tin with copper would only introduce some hundred µg g$^{-1}$ into the bronze, but not the lead concentrations observed in the *bronzetti*. This interpretation is supported by the collected data, which show no relationship between tin and lead.

### 4.4. Provenance of the tin

The above findings facilitate the tracing of the origin of the tin in the bronzes, as the tin isotope signature was not *per se* affected by the mixing of copper-based metals. Nevertheless, finding tangible evidence for the sources of the tin remains one of the greatest challenges of archaeometallurgical research. Recent studies have made considerable progress in answering provenance questions through the application of multiproxy approaches, but such procedures are often restricted to tin metal. For unalloyed tin, lead and tin isotopes as well as the trace elements can be used [60–62,131,132]. In the case of bronze, however, only the tin isotope composition is a useful tracer, as the lead isotope ratios and the chemical composition are usually dominated by the copper component. This limitation also applies to the studied figurines.

Although there are a number of tin minerals, cassiterite has since the early days of tin metallurgy been the chief tin ore mineral, whether exploited from alluvial deposits (placer) in open-pit mining or accessed from primary tin deposits by underground operations. In Sardinia, cassiterite is known from a few places in the Iglesiente-Sulcis region (Fig 3), but there are neither archaeological indications that these tin occurrences were used in the Bronze Age, nor are they suited for exploitation by primitive mining methods [18,97,133,120]. The cassiterite occurs mainly as a minor component and in tiny crystals hardly recognisable by ancient miners. However, since the possibility exists that the original cassiterite mineralisation was almost completely mined in ancient times, we must still consider the Sardinian tin occurrences as potential supplier.

Figure 18 shows the tin isotope composition of Sardinian cassiterite samples and compares it with that of the *bronzetti*. For comparison, the δSn data of the objects in Table 4 was corrected by an empirical factor of 0.025‰ u$^{-1}$ (= δSn$_{corr}$), determined in smelting experiments [135,136]. Without subtracting this factor, proper tracing would not be possible, because the tin isotope composition changes during the smelting process. It is clear from the diagram that the Sardinian cassiterite is significantly different from the tin in the bronzes. It is isotopically heavier compared to the artefacts, which have lower δSn values. The isotopically lighter tin in the bronzes can thus not derive from Sardinia, or at least not from Sardinia alone. Only if the tin smelted from Sardinian cassiterites had been mixed with isotopically lighter tin from other regions would the observed isotope composition have been obtained. However, there is currently no indication of this, also because isotopically heavy compositions, such as those observed in Sardinian cassiterites, are generally rare in European Late Bronze Age bronze and tin artefacts [60,61,137,134]. This is also the reason why the cassiterite from the Monte Valerio mine in Tuscany does not qualify as a source of the tin in the *bronzetti*: it stands out for even higher values than the Sardinian tin ores (Fig 18). In contrast, isotopically similar ores to the tin in the bronzes are widespread in other European deposits, especially in the large tin provinces of the Iberian Peninsula, of Cornwall/Devon, and of the Saxon-Bohemian Erzgebirge, but also in smaller deposits in the French Massif Central and farther away in Egypt. Since the transregional overlap in the tin isotope compositions is very large, the question of origin cannot be answered unequivocally.

If necessary, one could exclude smaller regions, or even individual mines, as suppliers of tin for the figurines, but this is of secondary importance for the argumentation here. On the one hand, it is more constructive to compare directly the

tin isotopy of the artefacts from the different sites. In doing so, one can conclude from the lower tin isotope data of the Abini bronzes (Fig 7d) that for their production tin from another source region or mine was used than for the *bronzetti* from the other sites. Conversely, tin from the same source region(s) was likely used for the bronzes from Santa Vittoria, the unknown site, and from Su Monte, given the generally similar isotopic spread. While this does still not allow for a precise identification of the tin's origin, it implies that the procurement of tin varied between sanctuaries, although some may have shared similar supply networks or consumption patterns.

On the other hand, it is revealing to include the tin isotope composition of Nuragic tin artefacts. Of the tin objects analysed so far, all have significantly lower δSn values ($\delta Sn_{corr}$ =−0.041 to +0.040‰ u$^{-1}$; n = 9) than Sardinian cassiterites ($\delta Sn$ = +0.086 to +0.128‰ u$^{-1}$; n = 8) [cf. [138,139]; unpublished data CEZA]. This is further evidence that external sources were preferred over local tin mineralisation. These could, for example, have been cassiterite from the Iberian Peninsula, which came to Sardinia along the same routes as the copper. The chemical composition of the Nuragic tin could be invoked as an indication of this, as it has similarly low indium concentrations (average ≤2.3 ± 0.9 µg g$^{-1}$) to the cassiterites from Iberia (average 1.6 ± 2.2 µg g$^{-1}$) and a Late Bronze Age tin object from Huelva (<3 µg g$^{-1}$) [62,135,140]. British cassiterite, for its part, has been found to be much richer in indium (28 ± 30 µg g$^{-1}$) [62], and thus is most likely not the source. However, there is still too little information and data available for cassiterites and tin artefacts throughout Eurasia, so these preliminary observations do not allow more than speculation at this time. The same is true for the tin in bun ingot MA-202494. Unlike the statuettes, its low tin concentration of 0.23% could well be explained with a natural origin from accessory cassiterite in polymetallic copper ores, occurring in Sardinia [97], but whether this is really the case remains an open question without further research.

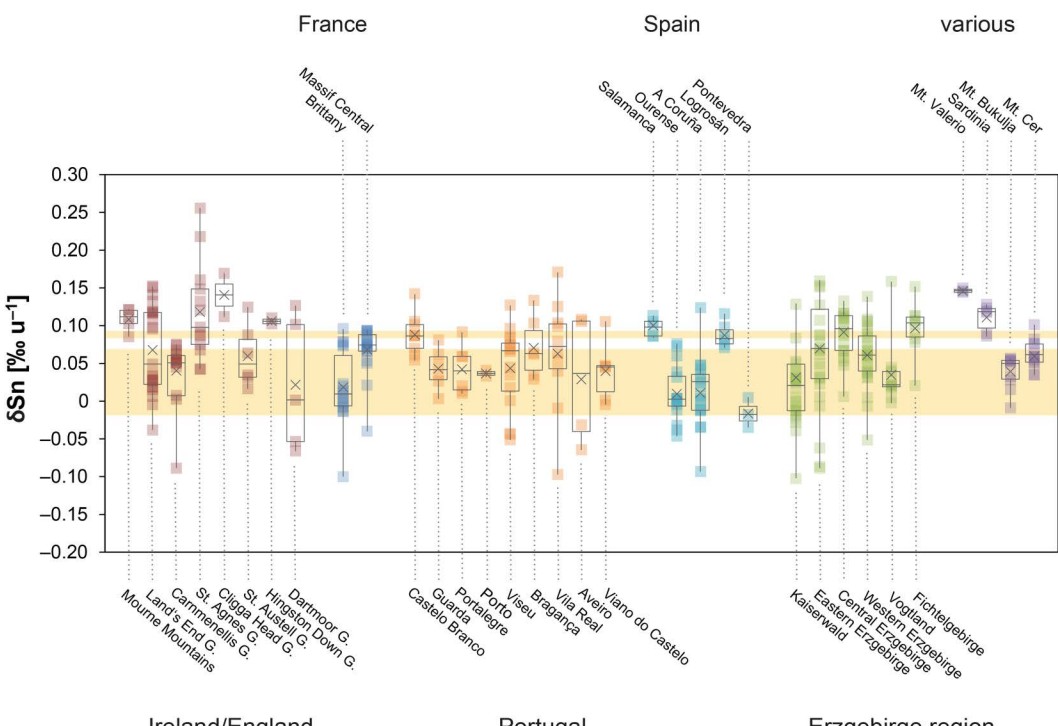

**Fig 18. Tin isotope values of the *bronzetti* compared with values from European tin provinces (diagrams: D Berger; data CEZA: [134]).**

## 5. Conclusions and synthesis

In this paper, fragments from a total of 48 bronze figurines (*bronzetti*) from the Nuragic sanctuaries of Santa Vittoria, Su Monte, and Abini, as well as an unspecified site in Sardinia have been investigated using a multiproxy approach. In addition to conventional chemical analysis with EDXRF and ICP-Q-MS and with lead isotope analysis, this approach comprises the rarely employed isotopic methods of copper and tin isotope analysis, and, for the first time, osmium isotope analysis. The findings from this integrated archaeometric investigation are very revealing and promising. The most significant result concerns the production of the figurines that involved copper from different sources, mixed either prior to or during the preparation of the bronze. The lead isotope ratios of the figurines from Santa Vittoria show an unnatural trend: they correlate, whether positively or negatively, with most trace elements, indicating an anthropogenic cause, with copper mixing as the most plausible explanation. From the inclusion of copper isotope values, and the absence of any observed proper mixing lines, it can be concluded that the Santa Vittoria figurines as well as some from Su Monte were not produced during a single mixing and casting event, but formed over an extended period in several mixing operations. For this, the craftspeople resorted to copper primarily from two distinct metal pools, as indicated by a clustering of data points in the lead isotope ratios and the consistent (though slightly differing) chemical and copper isotope data. The clusters can at the same time be perceived as the signatures of the original materials used for mixing. Thus, a complete mixing line (or rather a mixing belt) exists, including end members and mixtures. This greatly facilitates reconstruction of the origin of the copper, despite mixing.

While the metal of one of these end members points to ores in the Iglesiente-Sulcis district in Sardinia, most likely the Sa Duchessa mine, the other one was evidently produced from deposits in the Iberian Peninsula, likely situated in either the Alcudia valley or the Linares mining districts. In this regard, the highly radiogenic osmium isotope ratios are proof that Sardinian copper was indeed used as one metal pool, rather than copper from elsewhere being alloyed locally with lead from Sardinian deposits. This result is of great cultural-historical significance, as it clearly demonstrates that Sardinian copper mineralisation was exploited and used in local Nuragic metalwork. It had previously been argued that the Nuragic people made use of local ores, including those from the Sa Duchessa mine, but this was mainly based on the chemical and lead isotope systematics of artefacts [6,27,46]. With the highly radiogenic osmium isotope signatures as a putative characteristic feature of copper mineralisation in southwest Sardinia, we now have a very useful fingerprinting tool at hand, which is capable of confirming a Sardinian copper origin even if the lead isotope signature had potentially been altered by the deliberate addition of lead (in case of lead in high concentrations). It could even help to distinguish between Sardinian copper and copper from other regions with similar lead isotope systematics. Evidently, more fundamental research is necessary to understand the geochemical background and the osmium isotope systematics of the local Sardinian copper ores, as well as whether Iberian (and Levantine) copper deposits can be differentiated on these grounds. The latter aspect is even more topical because Iberia, the Levant, and Sardinia share major overlaps in the lead isotope ratios. This issue is especially problematic for some of the Su Monte *bronzetti*, which do not allow for clear conclusions about copper provenance.

Regarding both Santa Vittoria and the unknown location, an Iberian origin for the copper or copper-lead alloy used for part of the *bronzetti* is extremely likely, representing the second metal pool used for mixing. This could indicate that for a typical Nuragic object type of the early first millennium BCE, not only local, but also foreign metal sources were used. We lack sufficient archaeological information from the sites, and are thus currently not in a position to explain this ambiguity in metal sources. One possibility could be the craftspeople's awareness of variable copper qualities and their effects on casting, strength, and colours. Another possibility is that both foreign and local copper were reaching local markets and that Nuragic metallurgists simply picked what was immediately available or what was cheaper at the moment of purchase, without having particular preferences. This, for example, would explain why different batches of *bronzetti* from different sites have different compositions. It is likewise possible that local copper deposits might have become exhausted in this late Nuragic period, necessitating the search for non-local alternatives [46]. Whether this situation indeed applies, and

whether we are looking at a period in which a change of resources occurred, with copper from multiple sources used concurrently, is presently not possible to determine.

Furthermore, it is currently not possible to conclusively answer the question of whether the metal was mixed or recycled. Although similar, the intentions behind mixing and recycling are distinct. Nevertheless, both can lead to the same scientific results or relations in data. In the present case, however, recycling of existing metalwork appears a rather unlikely scenario, as no correlations are observed between any parameter and the tin isotope composition, correlations which would be expected if copper-tin alloys were being recycled. In the Final Bronze and Early Iron Age, on the other hand, bronze was omnipresent. Fragmentation of objects, especially rings, is common among the objects accumulated at the sanctuaries. Thus, recycling would be reasonable to infer, particularly for objects such as ornaments and figurines, for whose production and use material properties would be less essential than, for example, in weapons and tools.

This line of reasoning, in turn, has implications for the origin region of the tin. Since the tin isotope values of all analysed *bronzetti* exclude local Sardinian tin mineralisation, which consistently has higher values, the tin used for the figurines must have been imported. This conclusion aligns with geological arguments [18], which challenge the value of the Sardinian tin mineralisation for ancient metallurgists. The most plausible alternative origin is tin from the Iberian Peninsula, which probably travelled along the same sea routes as the copper for the *bronzetti*. Supporting this hypothesis are the solid evidence for contacts between the two cultural spheres during that time [5,141–144], as well as the trace-element composition of Bronze and Iron Age tin objects from Sardinia. All tin artefacts analysed so far are low in indium, which has recently been identified as a very useful parameter for tin provenance research [60,62]. Low concentrations speak in favour of Iberian tin ores rather than those from southwest Britain or the German–Czech Erzgebirge. The tin ores of the latter provinces were found to be richer in indium (especially Britain) than those of Iberia [62].

The metallurgical results confirm the strong connections already in evidence between Sardinia and the metalliferous southwest of the Iberian Peninsula in the early first millennium BCE. Both these interconnected regions stand out as hubs that were intermediaries between the Mediterranean and a wider Atlantic *oikumene* [145]. This connection, as revealed in the metals, is archaeologically echoed in the appropriation of similar-type weaponry wielded by mythical horned-helmet warrior figures in both places. There are also differences. Iberia lacks bronze figurines, instead promoting pictorial stone stelae presenting a very similar theme of warriors, other humans, and objects, together forming a sort of 'stelae community' as analogue to the '*bronzetti* community'. Stelae were typically erected and commemorated near copper lodes, while Sardinia preferred bronze figurines and giants in stone in the setting of sanctuaries [39]. The so-called gamma-hilted dagger (*stiletto*) is, conversely, an exclusively Sardinian brand in natural size and in miniature format among the symbolic components of *bronzetti* warriors [38].

These *bronzetti* and other late Nuragic objects and imports inform about a much larger Sardinian network in the early first millennium BCE, extending far beyond the Iberian link by hundreds of miles [31,38]. At this time, early Phoenician settlements were being founded along Sardinia's southwestern coastlands adjoining the Iglesiente metalliferous mountains. Significantly and certainly not a coincidence, the Mont'e Prama sanctuary locates in the same coastal region, flagging its iconic monumental sculptures whose form and innate ideas lean towards the Levant and even Egypt. In the same period, local metallurgy also adopts eastern Mediterranean models, especially vases and tripods [146]. These eastward-looking connections are absent or invisible in the present metallurgical results, which proved negative regarding Cypriot copper in the making of figurines.

This seems at odds with the numerous Cypriot oxhide ingots reported from Sardinia, but Cypriot copper was clearly not used to produce the analysed *bronzetti* (*cf.* S3 Appendix). In search of reasons for the absence of Cypriot copper, it is relevant to note that while oxhide ingots probably reached Sardinia in the Late Bronze Age (c. 1300–1100 BCE), they kept on circulating in fragments at least until the ninth century BCE, around the time the earliest *bronzetti* were made. A concomitant reason may be that Iberian and Sardinia's own sources mostly satisfied the needs of the Nuragic people for copper, especially those of the Iglisiente-Sulcis region, where the Sa Duchessa copper lode now confirms as the favourite.

If some oxhide ingots were still in circulation, they were not mixed nor recycled, possibly due to some sort of symbolic significance, or simply because they circulated as weight-regulated money, as bronze fragments likely did in the rest of Europe [147].

Even though many aspects of Nuragic metallurgical production are still not well understood, our results stimulate a preliminary discussion of the role of the Sardinian sanctuaries in the metallurgical train from the provision of raw material to the treasuring of the finished objects and hack metal. An important result of the multiproxy metallurgical investigation of *bronzetti* and bun ingots is indeed to have pinpointed the sanctuaries as an arena for handling and curating metals (the treasury function) and creating a demand for highly symbolic votive offerings (the *bronzetti* and the boat models). The sanctuaries were settings for disparate activities, including perhaps metallurgy [50,51,148,149]. By far the greatest number of metal objects from the Bronze Age discovered in Sardinia tie up with the sanctuaries, including the *bronzetti* and bun ingots analysed above. Three institutions surely existed and hypothetically they formed a triangular relationship: 1. *Mine* (e.g., Sa Duchessa) – 2. *Nuraghe village* (e.g., Sa Domu'e s'Orcu) – 3. *Sanctuary* (e.g., Matzanni) (Fig 19).

The operationalisation of this threefold network must have involved well-defined roles, procedures, and responsibilities. The above multiproxy investigation has indicated that specialised craftspeople were involved, and likely worked within the frame and auspices of the numerous sanctuaries known. This result is in accordance with the archaeological evidence. The sanctuaries were built upon flowing groundwater to channel and preserve the water of temples. In addition to the

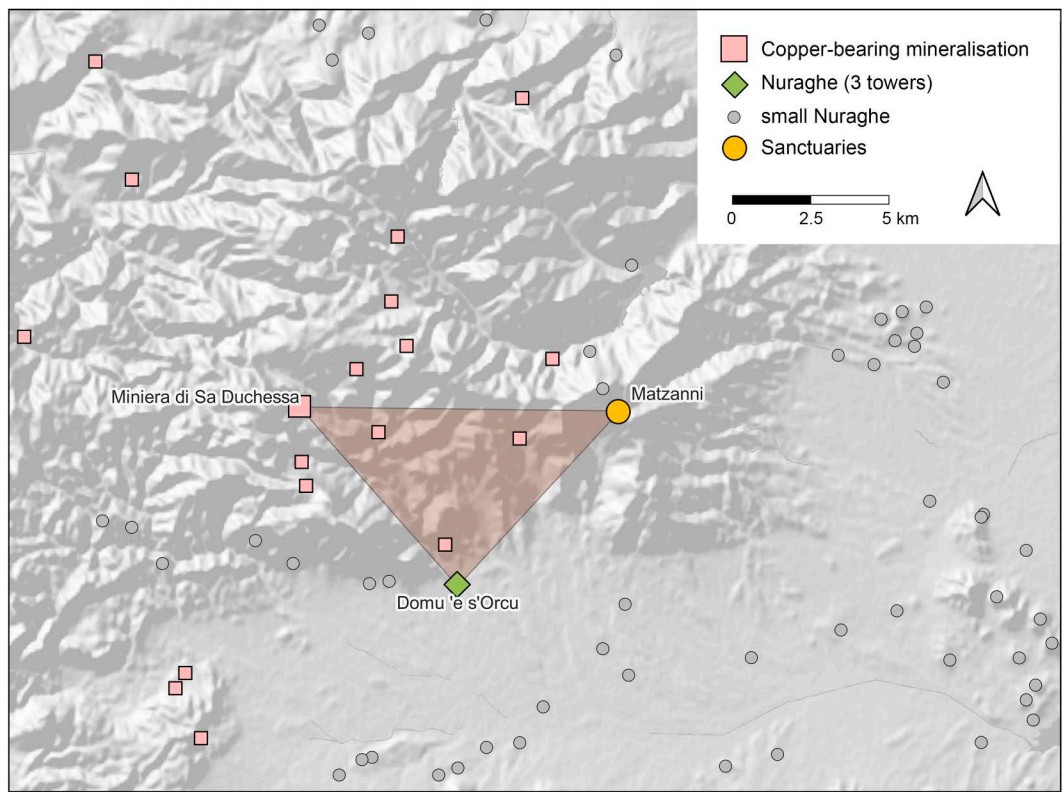

**Fig 19. Map showing the potential close relationship between the three 'institutions' of large nuraghe, sanctuary and copper mine; here exemplified by the Matzanni mountain sanctuary, the impressive three-towered Nuraghe Domu 'e s'Orcu and the Sa Duchessa copper mine.** Hypothetically, they formed a principal operational triangle in the Iglesiente-Sulcis region of SW Sardinia during the Final Bronze and Early Iron Age. The present study demonstrates that the Sa Duchessa mine was a primary source for the investigated *bronzetti*, along with Iberian copper (after [150]) (map: HW Nørgaard using Open Source QGIS software 3.20 Odense based on the ESRI shaded relief map, both CC BY4.0 license).

presence of water, these commemorative precincts are often erected proximate to organic fuel available in the vast areas of maki bushes that could be easily cut and used. Water and wood in excess, both key ingredients of pyrometallurgical processes, were available near the sanctuaries [148]. How the sanctuaries were managed daily is one of several questions that need more study. The just-started Metals & Giants research project has trained the scientific spotlight on micro-level histories, in this special case on and around the huge Matzanni mountain sanctuary located in the high ranges of the Iglesiente-Sulcis region [52]. The site is surrounded by mineralisations of copper, iron, and tin in addition to abundant lead and silver ores. Matzanni is located within walking distance of the nuraghe Sa Domu'e s'Orcu at Domusnovas as well as the nearby Sa Duchessa copper mine, which most likely delivered copper for the *bronzetti* analysed in this study (Fig 19). It is quite possible that Matzanni, as indicated by its size and dominant location at an altitude of 900 m, controlled the metallurgical activities in and around the precinct itself. Control might plausibly have been kinship-based, and directed from leading nuraghe villages.

Such micro-level history shapes and is shaped by the overriding geopolitical situation during the shadowy end of the Bronze Age and the emergence of the Iron Age in the early first millennium BCE. Around this time, Sardinia emerged as one of a number of geopolitical nodes in the greater Mediterranean, pulling together the networks of the west and the now post-palatial east in search of coveted metals. Southwest Sardinia and Iberia may still have attracted exogenous partners chasing copper and tin, but increasingly also lead/silver (*galena argentifera*) and iron. There is clearly a growing demand for silver among Cypriot and Levantine communities, and Sardinian argentiferous lead has been found at several sites there [104,105,107,151], hence clearly linking Sardinia to Aegean and Levantine traders.

## Sample information

All necessary permits were obtained for the described study from the Soprintendenza Archeologia, Belle Arti e Paesaggio per la Città Metropolitana di Cagliari e le Province di Oristano e Sud Sardegna, which complied with all relevant regulations.

## Supporting information

**S1 Appendix. Sample preparation and analytical methods.**
(PDF)

**S2 Table. Chemical and isotopic data of the figurines.** Chemical composition (in mass%), lead isotope ratios, osmium isotope ratios and osmium concentrations, copper isotope values (in ‰ relative ton NIST SRM 976) and tin isotope composition as individual isotopic values relative to NIST SRM 3161a reference material and Puratronic in-house reference material (in ‰ and as δSn in ‰ u$^{-1}$) of Sardinian bronzetti and ingots of the study (data: D Berger, G Brügmann, M Brauns B Höppner, N Lockhoff, CEZA).
(XLSX)

**S3 Appendix. The question of Cypriot copper for the production of the *bronzetti*.**
(PDF)

**S4 Appendix. Additional references.**
(PDF)

## Acknowledgments

We are very grateful for the permit by the Soprintendenza Archeologia, Belle Arti e Paesaggio per la Città Metropolitana di Cagliari e le Province di Oristano e Sud Sardegna to sample the *bronzetti* fragments and ingots, and to Dr C Pilo and Dr A Usai for help during sampling in the archaeological storerooms of Cagliari and Sorradile. We thank B Höppner, G

Brügmann, M Lockhoff and S Klaus, CEZA Mannheim, for sample preparation and for performing the chemical analyses with ICP-Q-MS and isotopic measurements with MC-ICP-MS.

## Author contributions

**Conceptualization:** Daniel Berger, Heide W. Nørgaard, Helle Vandkilde.

**Data curation:** Daniel Berger, Valentina Matta, Heide W. Nørgaard.

**Formal analysis:** Daniel Berger.

**Funding acquisition:** Daniel Berger, Gianfranca Salis, Mads K. Holst, Helle Vandkilde.

**Investigation:** Daniel Berger, Michael Brauns.

**Methodology:** Daniel Berger, Michael Brauns.

**Project administration:** Helle Vandkilde.

**Resources:** Valentina Matta, Heide W. Nørgaard, Gianfranca Salis.

**Supervision:** Helle Vandkilde.

**Validation:** Daniel Berger, Michael Brauns.

**Visualization:** Daniel Berger, Valentina Matta, Heide W. Nørgaard.

**Writing – original draft:** Daniel Berger, Valentina Matta.

**Writing – review & editing:** Daniel Berger, Valentina Matta, Nicola Ialongo, Heide W. Nørgaard, Gianfranca Salis, Michael Brauns, Helle Vandkilde.

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
