## [Decision Letter · Decision Letter 0]

23 Apr 2025

PONE-D-25-10968Multiproxy analysis unwraps origin and fabrication biographies of Sardinian figurines: On the trail of metal-driven interaction and mixing practises in the early first millennium BCEPLOS ONE

Dear Dr. Berger,

Thank you for submitting your manuscript to PLOS ONE. After careful consideration, we feel that it has merit but does not fully meet PLOS ONE’s publication criteria as it currently stands. Therefore, we invite you to submit a revised version of the manuscript that addresses the points raised during the review process.

We look forward to receiving your revised manuscript.

Kind regards,

Vanessa Forte, PhD

Academic Editor

PLOS ONE

Journal Requirements:

2. In your manuscript, please provide additional information regarding the specimens used in your study. Ensure that you have reported human remain specimen numbers and complete repository information, including museum name and geographic location.

For more information on PLOS ONE's requirements for paleontology and archeology research, see https://journals.plos.org/plosone/s/submission-guidelines#loc-paleontology-and-archaeology-research .

3. Thank you for stating the following in the Acknowledgments Section of your manuscript: [We are grateful to the Augustinus Foundation for funding the Metals & Giants project 2024–2027 (grant agreement 23–1869) and to the Danish Ministry of Culture FORM 2019 for funding analyses and the initial fieldwork at Matzanni, begun in 2019. Further funding for stable isotope analysis was obtained from the European Research Council through an Advanced Grant (grant agreement no. 323861). Osmium isotope analyses were made possible by funding resources from CEZA. The research presented is moreover indebted to Aarhus University Arts for funding V Matta’s PhD project, and to the research programme Material Culture and Heritage for covering costs of extra analyses. We are very grateful to Dr C Pilo and Dr A Usai for help during sampling in the archaeological storerooms of Cagliari and Sorradile (the Soprintendenza Archeologia, Belle Arti e Paesaggio per la Città Metropolitana di Cagliari e le Province di Oristano e Sud Sardegna). We thank B Höppner, G Brügmann, M Lockhoff and S Klaus for sample preparation and for performing the chemical analyses with ICP-Q-MS and isotopic measurements with MC-ICP-MS.]

Please remove any funding-related text from the manuscript and let us know how you would like to update your Funding Statement. Currently, your Funding Statement reads as follows: [Grant agreement 23–1869 to HV, MKH, GS. Augustinus Foundation funding the Metals & Giants project. https://augustinusfonden.dk/en

The foundation played no role in the study design, data collection and analysis, decision to publish, or preparation of the manuscript]

4. We note that there is identifying data in the Supporting Information file <S2 Table.xlsx>. Due to the inclusion of these potentially identifying data, we have removed this file from your file inventory. Prior to sharing human research participant data, authors should consult with an ethics committee to ensure data are shared in accordance with participant consent and all applicable local laws.

-Location data

Please remove or anonymize all personal information (ID), ensure that the data shared are in accordance with participant consent, and re-upload a fully anonymized data set. Please note that spreadsheet columns with personal information must be removed and not hidden as all hidden columns will appear in the published file.

5. We note that Figures 1,3 and 19 in your submission contain [map/satellite] images which may be copyrighted. All PLOS content is published under the Creative Commons Attribution License (CC BY 4.0), which means that the manuscript, images, and Supporting Information files will be freely available online, and any third party is permitted to access, download, copy, distribute, and use these materials in any way, even commercially, with proper attribution. For these reasons, we cannot publish previously copyrighted maps or satellite images created using proprietary data, such as Google software (Google Maps, Street View, and Earth). For more information, see our copyright guidelines: http://journals.plos.org/plosone/s/licenses-and-copyright.

1. You may seek permission from the original copyright holder of Figures 1,3 and 19 to publish the content specifically under the CC BY 4.0 license. 

Additional Editor CommentS:

Thank you for your submission and patience in receiving the reviews. The referees were all very positive regarding the paper and suggested minor revisions. I therefore return the paper to you under the category of minor revisions to give you the chance to reflect on the reviewer comments and make any necessary changes prior to final acceptance of the paper. Thank you for choosing Plos One.

Reviewers' comments:

Reviewer's Responses to Questions

**Comments to the Author**

1. Is the manuscript technically sound, and do the data support the conclusions?

Reviewer #1: Yes

Reviewer #2: Yes

2. Has the statistical analysis been performed appropriately and rigorously? 

Reviewer #1: Yes

Reviewer #2: Yes

3. Have the authors made all data underlying the findings in their manuscript fully available?

Reviewer #1: Yes

Reviewer #2: Yes

4. Is the manuscript presented in an intelligible fashion and written in standard English?

Reviewer #1: Yes

Reviewer #2: Yes

5. Review Comments to the Author

Reviewer #1: The study presents important new data that inform long-standing questions in the study of LBA - EIA Sardinia / central & eastern Mediterranean, metallurgical production and technological / communication networks. As such, the paper is a very welcome addition to the existing literature. I particularly enjoyed the suppl. mat. on Cypriot copper, which answers many of my questions that formed from reading the paper. I think a comment on the potential mixing of copper could / should be added in the main text, particularly considering recent results reported from the Balkans.

Although I am happy with the paper in its present form, i would like to invite the authors to a few considerations:

A comment on the possibility of a correlation between the reported mixing lines / ores and the typology / form / function of the analysed objects (I understand that they are mostly fragmens) in the text would be welcome.

Furthermore, I have some comments on figure 10: Fig. 10 only works if the population within each histogram is somehow normalized and, thus, made comparable to its lake-mirror histogram. As of now, any differences in the distribution could result from sampling bias. This would also fix the frequency issue in the Pb histograms. Importantly, all x axes need ticks and some indication of the binning / scale.

In a few occasions, where % with five decimals are give, possibly ppm values would be best reported.

Finally, I wonder if the long tables with the results should be best included in an xl file as suppl. mat. This would increase the flow / readability of the text and also make the data more easily reusable by future studies. Which leads me to ask: are these tables already duplicated in the 1st worksheet of S2 Table xl file?

Considering the above, I am very happy to see this paper published.

Reviewer #2: This is a technical analysis Sardinian metalwork dating to the early 1st millennium BC from the Nuragic sanctuaries of Santa Vittoria, Su Monte, and Abini. The two key aspects that stand out in terms of the significance of the study are the sophistication of the analytical approach to complex questions of metal provenance and mixing and the relative lack of comparable metallurgical analyses on the iconic object category - the iconic bronzetti. Given the central role of Sardinia in the Mediterranean metals trade in the early 1st millennium BC (and earlier) and the potential for sources spanning the Levant to Iberia, establishing the source of metals and the extent/nature of mixing/recycling is a challenge.

The paper is systematic and detailed in the analysis of the samples and admirably cautious in its interpretations. It nonetheless highlights persuasively that the copper is likely from the Sa Duchessa mine in Sardinia and the Alcudia valley or the Linares district in Iberia. However, the most useful contribution is the osmium isotope signature that demonstrates clearly an approach and result that shows the exploitation of Sardinian copper sources by local communities for the bronzetti. This should lay to rest a long running debate due to issues with lead isotope analyses and lead contamination and will be very useful for researchers in future. There is also a strong and useful analytical point in this assemblage on the lack of exploitation of Sardinian tin - another long running debate. The results evaluating mixing are certainly interesting though unsurprisingly not as definitive. It is however clear that Cypriot copper was not involved.

The authors are very well positioned indeed to lead this research and draw extensively and impressively on a wide range of scholarship, including their own recent and internationally significant research. They obtained the requisite permissions for sampling which should also be commended.

However, there needs to be a greater clarity and narrative flow for the reader in this paper which requires the authors to step back and consider what they want to say to the broader audience with this excellent paper. Otherwise there is a real potential for losing the reader. Currently, the introduction highlights that the paper addresses the debates over the local production of metal on Sardinia with the Cypriot alternative as a hypothesis. The analysis then not only addresses the debates on provenance - and highlights Iberia but goes into detail on questions of mixing/recycling/production sequences. The discussion addresses both of these and then rather abruptly provides a summary societal/landscape sanctuary model based on the very early results of the regional Sardinian project that the authors recently started. It is not always clear where the discussions over/references to Cyprus, the Levant, Iberia or elsewhere fit in and given that the main contribution is on Sardinia in the metals trade then this needs to be expanded and clarified. It currently feels if different sections where written by different authors with overlapping but slightly different purposes in mind for the paper.

6. PLOS authors have the option to publish the peer review history of their article (what does this mean? ). If published, this will include your full peer review and any attached files.

**Do you want your identity to be public for this peer review?** For information about this choice, including consent withdrawal, please see our Privacy Policy .

Reviewer #1: No

Reviewer #2: No

---

## [Author Response · Author response to Decision Letter 1]

12 Jun 2025

In the following, we reply to the editor and reviewer directly after their comments.

We marked our comments with: Comment of the authors: “…”

PONE-D-25-10968

Multiproxy analysis unwraps origin and fabrication biographies of Sardinian figurines: On the trail of metal-driven interaction and mixing practises in the early first millennium BCE

PLOS ONE

Dear Dr. Berger,

Thank you for submitting your manuscript to PLOS ONE. After careful consideration, we feel that it has merit but does not fully meet PLOS ONE’s publication criteria as it currently stands. Therefore, we invite you to submit a revised version of the manuscript that addresses the points raised during the review process.

We look forward to receiving your revised manuscript.

Kind regards,

Vanessa Forte, PhD

Academic Editor

PLOS ONE

Journal Requirements:

Comment of the authors: “Checked and, as far as we can assess it, our manuscript complies with PLOS ONE's style requirements.”

2. In your manuscript, please provide additional information regarding the specimens used in your study. Ensure that you have reported human remain specimen numbers and complete repository information, including museum name and geographic location.

Comment of the authors: “We have inserted the respective sentence in the manuscript in the chapter ‘Sample information’. The person who has permitted sampling is co-author of the paper (G Salis).”

For more information on PLOS ONE's requirements for paleontology and archeology research, see https://journals.plos.org/plosone/s/submission-guidelines#loc-paleontology-and-archaeology-research.

3. Thank you for stating the following in the Acknowledgments Section of your manuscript: [We are grateful to the Augustinus Foundation for funding the Metals & Giants project 2024–2027 (grant agreement 23–1869) and to the Danish Ministry of Culture FORM 2019 for funding analyses and the initial fieldwork at Matzanni, begun in 2019. Further funding for stable isotope analysis was obtained from the European Research Council through an Advanced Grant (grant agreement no. 323861). Osmium isotope analyses were made possible by funding resources from CEZA. The research presented is moreover indebted to Aarhus University Arts for funding V Matta’s PhD project, and to the research programme Material Culture and Heritage for covering costs of extra analyses. We are very grateful to Dr C Pilo and Dr A Usai for help during sampling in the archaeological storerooms of Cagliari and Sorradile (the Soprintendenza Archeologia, Belle Arti e Paesaggio per la Città Metropolitana di Cagliari e le Province di Oristano e Sud Sardegna). We thank B Höppner, G Brügmann, M Lockhoff and S Klaus for sample preparation and for performing the chemical analyses with ICP-Q-MS and isotopic measurements with MC-ICP-MS.]

Comment of the authors: “Deleted.”

Please remove any funding-related text from the manuscript and let us know how you would like to update your Funding Statement. Currently, your Funding Statement reads as follows: [Grant agreement 23–1869 to HV, MKH, GS. Augustinus Foundation funding the Metals & Giants project. https://augustinusfonden.dk/en

The foundation played no role in the study design, data collection and analysis, decision to publish, or preparation of the manuscript]

Comment of the authors: “We included the statement within the cover letter.”

4. We note that there is identifying data in the Supporting Information file <S2 Table.xlsx>. Due to the inclusion of these potentially identifying data, we have removed this file from your file inventory. Prior to sharing human research participant data, authors should consult with an ethics committee to ensure data are shared in accordance with participant consent and all applicable local laws.

-Location data

Please remove or anonymize all personal information (ID), ensure that the data shared are in accordance with participant consent, and re-upload a fully anonymized data set. Please note that spreadsheet columns with personal information must be removed and not hidden as all hidden columns will appear in the published file.

Comment of the authors: “Removed.”

5. We note that Figures 1,3 and 19 in your submission contain [map/satellite] images which may be copyrighted. All PLOS content is published under the Creative Commons Attribution License (CC BY 4.0), which means that the manuscript, images, and Supporting Information files will be freely available online, and any third party is permitted to access, download, copy, distribute, and use these materials in any way, even commercially, with proper attribution. For these reasons, we cannot publish previously copyrighted maps or satellite images created using proprietary data, such as Google software (Google Maps, Street View, and Earth). For more information, see our copyright guidelines: http://journals.plos.org/plosone/s/licenses-and-copyright.

Comment of the authors: “Maps in Figs 1 and 3 have been produced Open Source QGIS software 3.20 geo reference data from this source: https://www.sardegnageoportale.it/index.php?xsl=2420&s=40&v=9&c=14468&es=6603&na=1&n=10&esp=1&tb=14401

Map in Fig 19 have been produced Open Source QGIS software 3.20 based based on the ESRI shaded relief map.

All data there is published under the CC BY 4.0 license.

No special license request is necessary.”

1. You may seek permission from the original copyright holder of Figures 1,3 and 19 to publish the content specifically under the CC BY 4.0 license.

Comment of the authors: “We added such a phrase in the respective captions.”

Comment of the authors: “Checked. It is complete and correct.”

Additional Editor CommentS:

Thank you for your submission and patience in receiving the reviews. The referees were all very positive regarding the paper and suggested minor revisions. I therefore return the paper to you under the category of minor revisions to give you the chance to reflect on the reviewer comments and make any necessary changes prior to final acceptance of the paper. Thank you for choosing Plos One.

Reviewers' comments:

Reviewer's Responses to Questions

Comments to the Reviewers

Comment of the authors: “Dear reviewers, thanks for your time and valuable input regarding our manuscript. We hope to have addressed all the points raised appropriately. In the following we reply to your points directly after your lines.”

Comments to the Author

1. Is the manuscript technically sound, and do the data support the conclusions?

Reviewer #1: Yes

Reviewer #2: Yes

2. Has the statistical analysis been performed appropriately and rigorously?

Reviewer #1: Yes

Reviewer #2: Yes

3. Have the authors made all data underlying the findings in their manuscript fully available?

Reviewer #1: Yes

Reviewer #2: Yes

4. Is the manuscript presented in an intelligible fashion and written in standard English?

Reviewer #1: Yes

Reviewer #2: Yes

5. Review Comments to the Author

Reviewer #1: The study presents important new data that inform long

---

## [Editor Report · Decision Letter 1]

29 Jun 2025

Multiproxy analysis unwraps origin and fabrication biographies of Sardinian figurines: On the trail of metal-driven interaction and mixing practises in the early first millennium BCE

PONE-D-25-10968R1

Dear Dr. Berger

We’re pleased to inform you that your manuscript has been judged scientifically suitable for publication and will be formally accepted for publication once it meets all outstanding technical requirements.

Kind regards,

Vanessa Forte, PhD

Academic Editor

PLOS ONE